# From Narratives to Numbers: Valid Inference Using Language Model Predictions From Verbal Autopsy Narratives

**Shuxian Fan** [*1]  **Adam Visokay** [*1]  **Kentaro Hoffman**[1]  **Stephen Salerno**[2]
**Li Liu**[3]  **Jeffrey T. Leek**[2]  **Tyler H. McCormick**[1]

[1] University of Washington   [2] Fred Hutchinson Cancer Center
[3]Johns Hopkins University

## Abstract

In settings where most deaths occur outside the healthcare system, verbal autopsies (VAs) are a common tool to monitor trends in causes of death (COD). VAs are interviews with a surviving caregiver or relative that are used to predict the decedent's COD. Turning VAs into actionable insights for researchers and policymakers requires two steps (i) predicting likely COD using the VA interview and (ii) performing inference with predicted CODs (e.g. modeling the breakdown of causes by demographic factors using a sample of deaths). In this paper, we develop a method for valid inference using outcomes (in our case COD) predicted from free-form text using state-of-the-art NLP techniques. This method, which we call `multiPPI++`, extends recent work in "prediction-powered inference" to multinomial classification. We leverage a suite of NLP techniques for COD prediction and, through empirical analysis of VA data, demonstrate the effectiveness of our approach in handling transportability issues. `multiPPI++` recovers ground truth estimates, regardless of which NLP model produced predictions and regardless of whether they were produced by a more accurate predictor like GPT-4-32k or a less accurate predictor like KNN. Our findings demonstrate the practical importance of inference correction for public health decision-making and suggests that if inference tasks are the end goal, having a small amount of contextually relevant, high quality labeled data is essential regardless of the NLP algorithm.

## 1 Introduction

Verbal Autopsies (VAs) are a pivotal tool in public health research, particularly in contexts where access to formal medical records is limited (Soleman et al., 2006; Thomas et al., 2018). VA serves as a method to ascertain causes of death (COD) by collecting information from relatives or witnesses in circumstances necessitated by resource constraints, remote locations, or inadequate healthcare infrastructure (Baqui et al., 2006; Byass et al., 2012; Wahab et al., 2017; Blanco et al., 2021).VA interviews include a structured yes/no questionnaire and an open text narrative where the interviewees provide additional information about the illness or death in their own words.

After the interview, a COD is assigned either by clinician review or, more commonly, by using statistical algorithms. Finally, since financial and logistical constraints prevent VAs from being performed on all deaths, researchers and policymakers use statistical tools to summarize patterns in COD (e.g. the breakdown of infectious disease deaths by age, race, sex, etc).

We propose a method for valid inference using the open text VA narratives, summarized in Figure 1. To do this, we address two pressing challenges. First, we use text narratives

---
*Denotes first authorship

exclusively to classify COD. Structured VA interviews are long (typically around an hour) and emotionally taxing for respondents (Surek-Clark, 2020). Narratives provide an opportunity for respondents to describe the circumstances around the death of someone they knew well in their own words, without needing to translate clinical jargon or answer irrelevant questions. Using narratives alone could also dramatically reduce the length of the interview, both minimizing impact on the respondent and allowing enumerators to collect more VAs. Recent work (Danso et al., 2013; Blanco et al., 2021; Manaka et al., 2022; Cejudo et al., 2023) has begun exploring the potential of natural language processing (NLP) for COD classification in VA. Our work furthers this discussion by leveraging state-of-the-art NLP tools. Also, unlike prior NLP work which focuses on *predictive performance* (Naradowsky et al., 2012; Liu et al., 2019; Ye et al., 2021; Peskoff & Stewart, 2023), our goal is to utilize these predictions in *statistical inference* to understand patterns in COD.

Second, we propose a method for inference that corrects for misclassification in CODs predicted by NLP. Training and evaluating COD prediction algorithms is extremely challenging (Murray et al., 2011; Fottrell & Byass, 2010; Murray et al., 2014; Clark et al., 2018; Li et al., 2020), in part because building training sets with reliably labeled CODs is taxing. Clinicians could review VAs and assign causes, but this diverts critical resources from patient care and, since most deaths occur outside healthcare facilities, a physical autopsy is impossible. Hence, it is appealing to create a larger training set by combining the limited number of labeled VAs in one domain with other labeled VAs from different domains (e.g. locations). However, the diversity of cultural practices, linguistic nuances, and variations in interview techniques (not to mention potential biological factors) means that an algorithm trained on labeled VAs in one context may perform poorly in another. Our inferential methods must account for additional uncertainty that arises when most COD labels are predicted, not known. We also need to adjust for transportability so that VAs labeled in other contexts can be incorporated effectively into the training set. To do this, we extend prediction-powered inference (PPI++; Angelopoulos et al., 2023a;b) to settings of multinomial classification, showcasing its adaptability to a broader range of prediction problems. We propose an estimator, called multiPPI++, used to draw valid statistical inference using predictions from any arbitrary black-box machine learning model given access to a small subset of ground truth labels for multinomial data.

**The Workflow for Valid Inference Using multiPPI++ for VA Narratives.**

Figure 1: Overview of multiPPI++ correction. Ground truth labels and predicted labels are used separately to perform the same inference task in Domain A. We use the difference between these estimates as a correction factor in Domain B where ground truth labels are not available.

## 2 Population Health Metrics Research Consortium Narratives

The Population Health Metrics Research Consortium (PHMRC) dataset was collected in 2005 to provide a high quality, standardized dataset that includes ground truth COD labels from traditional autopsies, structured questionnaire responses, and free-text open narrative responses (Murray et al., 2011). It is one of a very small number of VA datasets with clinically validated causes. These data were curated from six sites across four countries: (*Andhra Pradesh and Uttar Pradesh, India; Bohol, Philippines; Dar es Salaam and Pemba Island, Tanzania; and Mexico City, Mexico*). An example PHMRC narrative follows: *"the deceased had been burnt and had lost mental balance and died within 1.5 hours of the accident"* (Ground Truth from Traditional Autopsy: **Fires**).

We focus our analysis on adult deaths, excluding those under 12 years of age, in the de-identified narratives released in Flaxman et al. (2018). CODs are categorized into five broader groups: non-communicable diseases, communicable diseases, external causes, maternal causes, and AIDS or tuberculosis (AIDs-TB), based on International Classification of Diseases Tenth Revision (ICD-10) codes (McCormick et al., 2016). See Appendix A.1 for the full mapping from ICD-10 codes to COD labels.

Under these five broad classes, there still remains considerable heterogeneity in the COD distribution between sites. Figure 2 illustrates this phenomenon. While "non-communicable" diseases (e.g., various cancers, COPD, stroke) are the most common COD across all six sites, the second most common cause is site-specific. Namely, two sites identify communicable diseases (e.g., malaria, pneumonia) as the second most common COD, while external causes (e.g., suicide, homicide, traffic accidents) are reported in three sites and AIDs-TB in one site. In addition, the difference in rates between the first and second most common CODs varies considerably, with, for example, a 5:4 ratio in Pemba versus a 6:1 ratio in Mexico.

These observations suggest that while information from one site may provide some insights into another, caution is warranted when extrapolating NLP model performance from one location to all others. In practice, this lack of transportability between sites poses a significant public health challenge and is well-documented in the VA literature on predicting COD (e.g., see McCormick et al., 2016). However, without a means of addressing this issue for downstream statistical inference, it may be necessary to establish COD data collection systems at each new site of interest, an endeavor that would likely prove prohibitively expensive. We address this explicitly in Section 3.2.

**Frequency Distribution of COD by Site**

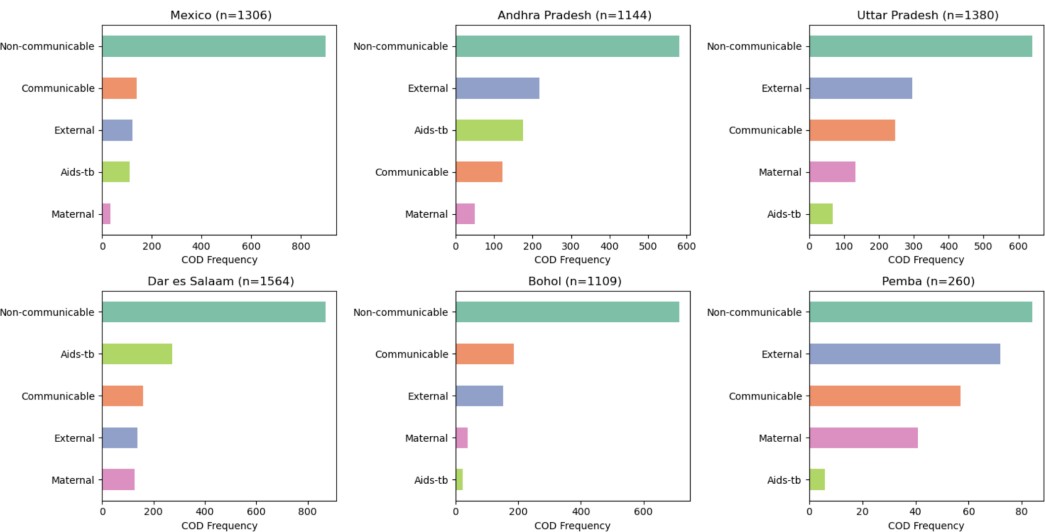

Figure 2: While non-communicable disease is the most common COD in each site, relative prevalence of each COD varies considerably.

## 3 Proposed Analytic Workflow

### 3.1 NLP for VA Narratives

The first aim of this work is to assess the predictive performance of contemporary NLP methods for classifying COD from free-text narratives, rather than structured interviews. To predict COD from the free-text narratives, we analyze a range of NLP techniques, including various methods based on a bag-of-words (BoW) representation and more sophisticated pre-trained models such as Bidirectional Encoder Representations from Transformers (BERT; Devlin et al., 2018) and large language models (LLMs) such as the recently popular Generative Pre-trained Transformer (GPT model; Lai et al., 2023).

To generate BoW representations of VA narratives, we employed `scikit-learn`'s `CountVectorizer`, `LabelEncoder`, and `TfidfVectorizer` (Kramer & Kramer, 2016; Pedregosa et al., 2011), as well as `nltk`'s `word_tokenize` (Millstein, 2020) for data pre-processing. After processing, the initial token count of $556,713$ was reduced to $218,804$. We then applied three supervised classifiers to predict COD using the BoW representations and ground truth cause of death labels: Support Vector Machine (SVM), K Nearest Neighbors (KNN), and Naive Bayes (NB). SVM was implemented with a third-degree linear kernel and $C = 1.0$, while KNN used cosine distance and 9 nearest neighbors, chosen by incrementing from $K = 3$ until accuracy saturation. The BERT$_{BASE}$ encoder was configured with five transformer layers, to include two input layers of size 256, a main layer of size 768, an intermediate layer of size 512, and a fine-tuning output layer of size 5. As the VA narratives in our data were de-identified, we were able to further assess the predictive performance of OpenAI's GPT models (GPT-3.5, GPT-3.5-turbo, GPT-4, and GPT-4-32K Biswas, 2023). After exploratory study, we selected GPT-4-32K for its larger context window. We used a zero-shot prompt (see Figure 3) with a temperature set to zero for all tasks. For all of the methods, we split the data into an 80% training set and 20% testing set, where we held out the true COD labels in the testing set to assess each method's predictive accuracy and F1-score. For BERT$_{BASE}$, training was conducted over two epochs.

We faced a series of challenges in engineering the LM prompt, namely, coercing the LM to consistently return outputs constrained to our exact list of output classes [communicable, non-communicable, external, maternal, aids-tb]. Earlier iterations of the prompt yielded responses such as "the patient died due to external causes." While perhaps semantically correct, validating outputs such as this at scale is intractable and would require additional steps (regex, fuzzy string matching, etc.). Instead, we found that including our desired output classes between `<option>` html tags and explicitly asking the LM to return only a choice from the list of options in the prompt improved results considerably. Future work may include few shot prompting with select domain-specific examples or retrieval augmented generation from a larger database of narratives.

### 3.2 Valid Statistical Inference with `multiPPI++` Correction

The second aim of this work addresses the challenge of conducting valid downstream inference on covariates associated with predicted COD labels. As long as the NLP model is not always 100% accurate, uncritical inference using NLP predictions will always contain bias, which we must correct for before trusting the results. Note that while the PHMRC dataset is complete with ground truth COD labels from traditional, physical autopsies, this is often not practical due to resource constraints. Particularly in vulnerable areas without healthcare infrastructure, COD is abstracted solely from VA without access to means of verification. Recent works have shown that reifying algorithmically-derived outcomes (here, COD) for downstream inference can lead to biased coefficient estimates and anti-conservative inference (Wang et al., 2020; Angelopoulos et al., 2023a;b; Miao et al., 2023; Egami et al., 2023; Ogburn et al., 2021; Hoffman et al., 2024).

To address this, we employ a recent technique known as prediction-powered inference (`PPI`; Angelopoulos et al., 2023a) and its extension, `PPI++` (Angelopoulos et al., 2023b) and derive its appropriate form in the context of multiclass logistic regression. These methods operate by using a small subset of ground truth COD labels to *rectify* the coefficient estimates and

**GPT-4 Zero-Shot Prompt Used for COD Prediction**

Figure 3: The zero-shot prompt explicitly tags the VA narrative, provides minimal context for each COD label, lists an explicit set of COD options to coerce a constrained output, and provides direct instructions pointing to the <narrative><label><option> tags.

standard errors from a regression based on predicted COD labels. To that end, assume that our data are comprised of $n$ *labeled* ($l$) observations, $\mathcal{L} = \{(Y_{l,i}, \hat{Y}^f_{l,i}, X_{l,i}, Z_{l,i}); i = 1, \ldots, n\}$, and $N$ *unlabeled* ($u$) observations, $\mathcal{U} = \{(\hat{Y}^f_{u,i}, X_{u,i}, Z_{u,i}); i = n+1, \ldots, n+N\}$, where $Y$ denotes the true cause of death, $\hat{Y}^f$ denotes the cause of death predicted from an NLP function, $f(Z)$, $X$ denotes the covariates of interest, and $Z$ denotes the VA text narratives. In statistical inference, one can frame estimating statistical parameters, $\theta$, as minimizing an objective function of the form:

$$\theta^{\textit{ Ground Truth}} = \arg\min_{\theta \in \mathbb{R}^d} \mathbb{E}[l_\theta(X, Y)], \tag{1}$$

where $l_\theta : \mathcal{X} \times \mathcal{Y} \to \mathbb{R}$ is a convex loss function in $\theta \in \mathbb{R}^d$. In our case, the loss function $l_\theta(X, Y)$ is defined as the negative log conditional likelihood of the multinomial logistic regression. In practice, replacing the observed COD, $Y$, with $\hat{Y}^f$ yields what we refer to as a *Naive* estimate:

$$\theta^{Naive} = \arg\min_{\theta \in \mathbb{R}^d} \mathbb{E}[l_\theta(X, \hat{Y}^f)]. \tag{2}$$

The Naive estimate relies heavily on the assumption that the NLP model's predictions closely resemble the true CODs.

PPI++, and our extension, multiPPI++, seeks to utilize all available information from the unlabeled and labeled data, by minimizing the following objective function:

$$\theta_\lambda^{PPI++} = \arg\min_{\theta \in \mathbb{R}^d} \left\{ \mathbb{E}[l_\theta(X_l, Y_l)] + \lambda \left( \mathbb{E}[l_\theta(X_u, \hat{Y}_u^f)] - \mathbb{E}[l_\theta(X_l, \hat{Y}_l^f)] \right) \right\}, \quad (3)$$

where $\lambda \in [0, 1]$ is a tuning parameter which controls how much weight is placed on using the real labels versus the predicted labels. Note that this estimator combines the ground truth estimator from the labeled subset, which is statistically valid but may lack sufficient sample size, with the Naive estimator, which provides additional information and data from the unlabeled subset.

We extend the application of the PPI++ approach from its logistic regression example to the multinomial case. Angelopoulos et al. (2023b) mention this is possible, but do not provide details of the derivation, and their model has overparameterization issues. We instead derive a different parameterization that avoids identifiability issues, and we complete the derivation of PPI++ correction for multiclass logistic regression. We refer to this variant as "multiPPI++", where, for the multinomial classification problem with $K$ classes and outcomes $Y_i \in \{0, \ldots, K-1\}$, our loss function takes the form

$$l_\theta(X, Y) = -\frac{1}{n} \sum_{i=1}^{n} X_i^T \theta_{Y_i} + \log \left( \sum_{k=1}^{K-1} e^{X_i^T \theta_k} \right).$$

We present the multiPPI++ adjustment procedures in Algorithm 1. See Appendix A.2 and A.3 for complete derivations.

## 4 Inference with COD Predicted from VA Narratives

### 4.1 Experimental Setup

To study the performance of the proposed methods, we designed an experiment in which we synthetically removed ground truth COD labels from the PHMRC dataset and replaced them with predicted COD from the NLP methods detailed above. Specifically, for each of the the six sites in the PHMRC dataset, we trained the NLP methods on the other five sites and used the resulting models to predict COD in the sixth site (see Figure 4).

We compared the predicted outcomes to the withheld, true outcomes for the sixth site in terms of predicted accuracy and F1-score. We then carried forward these predictions to a downstream inferential model in which we regressed site-specific COD on the age of the decedent, to illustrate the performance of the multiPPI++ estimator. For each site, we retained 20% of the labeled outcomes and replaced the remaining 80% with the NLP predictions, to mirror the *labeled* and *unlabeled* subsets of the data necessary for the multiPPI++ approach. We compared the results of the proposed estimator to two additional estimators: (i) the 'ground truth' estimator, which utilized the full set of true COD labels, and (ii) the 'Naive' estimator, which used only the predicted COD, treating them as if they were observed. Note that, in many practical settings, the ground truth estimator is not feasible, but it is included here as a point of reference. Further,

due to the variability in data between sites and the fact that the NLP model was not trained on this particular site, we anticipate that the regression coefficients estimated using the NLP-predicted COD will differ from those estimated using the true COD.

We exemplify one representative set of experiments in which the Uttar Pradesh site is excluded from the NLP model training and used as the validation site to assess the NLP models' predictive performance and the downstream inferential results.

### 4.2 NLP Prediction Results

First, using the "leave one site out" procedure described in Figure 4, we trained each NLP model to predict COD from narratives on five sites, with the sixth site reserved as a validation set to use for assessing the methods. As evidenced by the COD distribution

**"Leave One Out" Synthetic VA Transportability Experiment**

Figure 4: The classic and BERT models are trained with VA narratives from five other sites and used to predict COD from Uttar Pradesh narratives. The zero shot GPT-4 model predicts COD without any site specific narrative fine-tuning. Ground truth, Naive and `multiPPI++` corrected inference is performed using these Uttar Pradesh predictions.

displayed in Figure 2, we are dealing with considerable class imbalance within and between sites. We address this in a sensitivity analysis (see Figures 29-54 in A.6) where we train each model on a stratified sample with an 80/20 split for each class to ensure an equal proportion of training examples for each class. The results from this analysis are not substantively different from our main results where we do not account for the class imbalance in our modeling. Alternative resampling methods such as SMOTE, ADASYN or Tomek Links (Chawla et al., 2002; He et al., 2008; Tomek, 1976) could be applied to correct for the imbalanced distribution of COD. However, even in the absence of such considerations, we show that multiPPI++ still obtains valid estimates in downstream inference using predictions from imperfect NLP modeling. This extensibility of our approach is one of the advantages of the multiPPI++.

We focus our subsequent analysis one representative site, Uttar Pradesh, but share results from the other sites in the Appendix. These results for Uttar Pradesh are summarized in Table 1 and Figure 5, which contain the four models with highest accuracy. All models scored between 0.60-0.75 accurate, with GPT-4 yielding the highest accuracy (0.75). F1-scores were more uniform, with all but KNN getting F1-scores between 0.71-0.74. KNN scored 0.66. These results seem to point at GPT-4 either being slightly better or nearly equivalent to the other NLP methods we tested. GPT-4's predictions matched the performance of the best examples from the VA literature, despite using only using only the VA text narratives and not the structured questionnaire (James et al., 2011; Murray et al., 2014; Serina et al., 2015).

| Metrics | BoW w/ SVM | BoW w/ KNN | BoW w/ NB | BERT | GPT-4 |
|---------|------------|------------|-----------|------|-------|
| Accuracy | 0.68 | 0.63 | 0.60 | 0.55 | 0.75 |
| F1-Score | 0.74 | 0.66 | 0.72 | 0.71 | 0.73 |

Table 1: GPT-4 is far more accurate than competing NLP models and is among the best in terms of F1-score. Using only text narratives, GPT-4 performs on par with the best predictions in the VA literature, which are produced with much richer data in the form of structured questionnaires.

A unique feature of LLMs compared to other prediction algorithms is that they have the capacity, and in fact the tendency, to output class labels that do not exist in the ground labels. In our analysis, GPT-4 predicted 1503 out of 6763 COD labels as "unclassified". Upon manual review, these 1503 narratives contained no useful information for COD inference. The modal "unclassified" narrative was "the interviewee thanked the interviewer" (n=178). While the accuracies appear similar, GPT-4 correctly identified a flaw in the data cleaning in the PHMRC dataset, which suggest that the accuracies among the other methods may be overinflated. For each of the combination of six sites and five NLP model, we generated predictions and used this to generate regression coefficients.

For the sake of brevity, we present the four models with the best performance for the Uttar Pradesh site, NB, KNN, SVM, and GPT-4. Figure 6 displays point estimates and 95% confidence intervals for the ground truth, Naive and `multiPPI++` corrected inference.

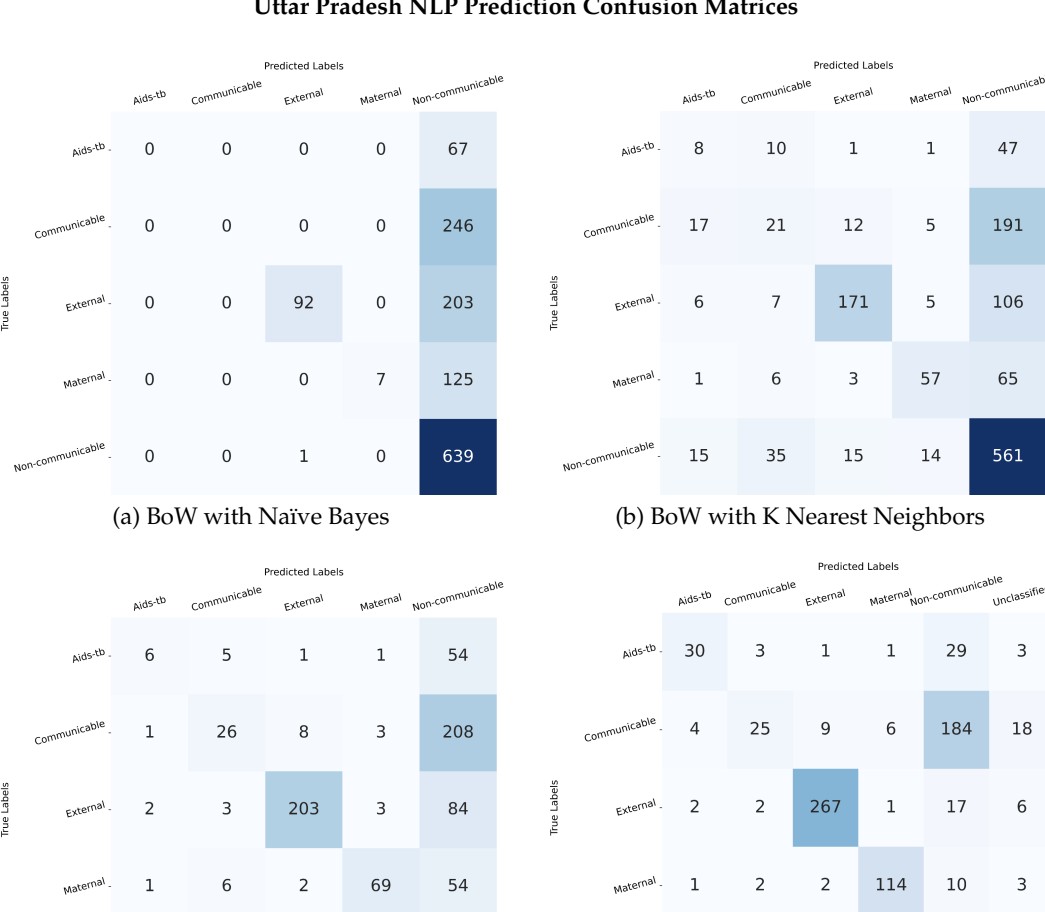

Figure 5: For VA narratives from Uttar Pradesh, most of the COD misclassifications are assigned non-communicable COD label. Naive Bayes mostly predicts non-communicable and achieves 0.60 accuracy in-part because non-communicable is overwhelmingly the most common ground truth COD.

## 4.3 Inferential Model Results

In the case of NB and SVM, which had some of the lower F1-scores, there is a large discrepancy between the ground truth and the predicted values, demonstrating that replacing 80% of the causes of death with predicted causes of death yields significantly different regression coefficients for the age of the decedent. `multiPPI++` correction rectifies estimates back to baseline, demonstrating its ability to account for (i) model inaccuracy and (ii) transportability bias. Interestingly, we observe that the Naive regression coefficients from KNN and GPT-4 are not as distant from the ground truth as the other two methods, even though KNN had the worst F1-score. Figure 7 shows the association between F1-score and `multiPPI++` $\lambda$, which also fail to show any significant association. This implies that F1-score alone may not be a good predictor of the quality of downstream statistical inference.

As for the widths of the confidence intervals, we observe that ground truth naturally had the thinnest, as it had the largest sample size and used real data points. On the other hand, the Naive estimate and `multiPPI++` correction show similar confidence interval widths. Since the widths of the confidence interval for `multiPPI++` is the sum of the widths of the Naive interval plus the width of the correction term, this seems to indicate that performing the `multiPPI++` correction does not seem to add a significant amount of uncertainty. This demonstrates that it can correct for transportability bias without much additional information cost. This is further evidenced by the seeming lack of association between the tuning parameter $\lambda$, which weights the use of the predicted outcomes in the downstream inferential model and the accuracy of the predictions. As $\lambda$ is a function of both the outcome and the associated features, we conclude that better predicted outcomes carry no additional useful information for parameter estimation.

Figure 6 demonstrates that multiPPI++ improves the accuracy of statistical estimates using a post-facto correction term estimated using relatively small amounts of labeled data. As this is applied after model predictions are made, multiPPI++ does not improve or degrade prediction accuracy. This procedure provides a correction for the biased statistical inference parameters and the deflated uncertainty intervals around them, given NLP produced estimates of likely COD from VA narrative predictions. This is a nuanced yet important distinction as these downstream statistical inference parameters are what describe the relationship between covariates, such as age, and cause of death distributions. This same intuition extends to other domains where the end goal is to use inference to guide decision making where NLP predictions are used in place of ground truth labels.

**Inferential Experiment Results**

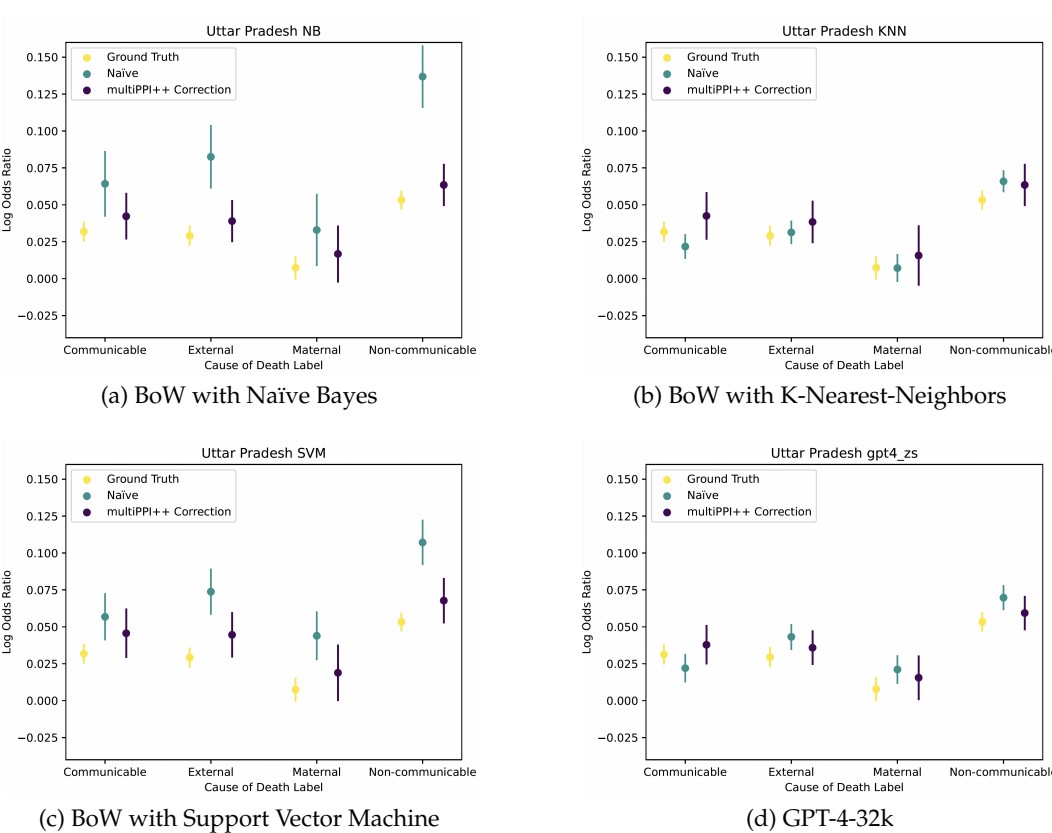

(a) BoW with Naïve Bayes

(b) BoW with K-Nearest-Neighbors

(c) BoW with Support Vector Machine

(d) GPT-4-32k

Figure 6: Ground truth, Naive estimation, and `multiPPI++` correction of estimation of log odds ratios for multinomial regression for age in Uttar Pradesh. `multiPPI++` correction both recovers point estimates and reflects the increased uncertainty corresponding to the use of predicted COD rather than known COD with wider confidence intervals.

**No Apparent Association Between F1-scores and multiPP++ $\lambda$'s**

Figure 7: There appears no strong global relationship between `multiPPI++` $\lambda$ and F1-score, though associations do emerge when subsetting by site, hinting at a Yule-Simpson effect.

## 5  Discussion

This paper uses a variety of NLP models to predict COD using VAs and, subsequently, develops a statistical method for valid inference using these predicted CODs. This research not only contributes to the advancement of predictive modeling in public health, but also underscores the significance of parameter interpretation in guiding decisions within diverse and dynamic health scenarios.

Based on our research, there are several points that we believe are worthy of further investigation. First, we noticed that better predictive accuracy in the COD classifications did not necessarily translate to improved parameter estimation when used as a generative model for downstream statistical inference, at least not under the accuracy ranges examined in this paper. In situations where the user is ultimately interested in accurate parameter estimation, as is often the case in public health, care must be taken to differentiate between the most accurate model for prediction versus the most accurate model for parameter estimation, as it is possible that very cheap models such as KNN can be as useful as more expensive, newer models like GPT-4. For example, the cost to train and predict COD with BoW representations and KNN, NB and SVM was trivial, whereas performing the same task with GPT-4 cost $3000USD.

Second, the extreme class imbalance in COD drives many of our prediction results. While it is not possible in practice to directly balance across CODs, there are broader design questions that could substantially improve inference. How, for example, should a researcher spend their very limited budget to label relevant VAs? And are there other ways of categorizing deaths that provide richer information than the five categories we use? Third, there is evidence that NLP techniques such as large language models perform worse in non-English languages Navigli et al. (2023); Lai et al. (2023); Hirschberg & Manning (2015), which is a major consideration since VAs are common in non-English settings. This could introduce bias to potential VA inference in other languages (Surek-Clark, 2020). This is a limitation of our simulation study, since it does not account for differential translation quality across languages. Future work might instead leverage NLP models specifically pre-trained for multiple languages or medical text (Peskoff et al., 2021; Singhal et al., 2023).

## Acknowledgements

We thank the University of Washington eScience institute for granting us Microsoft Azure computing credits used to produce all of our NLP predictions. We also thank Aidan Andronicos, Robert Fatland, Musashi Hinck, Eddie Hock, Jesse Peirce, Denis Peskoff, Nels Schimek, Sander Schulhoff, and Brandon Stewart for their help.

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

# A Appendix

## A.1 ICD-10 COD Classification

| Mapping 34 PHMRC All-Cause Mortality Labels to Five Broad COD Labels | | |
|---|---|---|
| **ICD-10 Code** | **All-Cause Mortality Label** | **Broad COD Label** |
| K74 | cirrhosis | non-communicable |
| G40 | epilepsy | non-communicable |
| J18 | pneumonia | communicable |
| J44 | copd | non-communicable |
| I21 | acute myocardial infarction | non-communicable |
| X00-X09 | fires | external |
| N17-N19 | renal failure | non-communicable |
| C34 | lung cancer | non-communicable |
| O00-O99 | maternal | maternal |
| W65-W74 | drowning | external |
| I00-I99 | other cardiovascular diseases | non-communicable |
| B20-B24 | aids | aids-tb |
| Varies | other non-communicable diseases | non-communicable |
| W00-W19 | falls | external |
| V01-V99 | road traffic | external |
| E10-E14 | diabetes | non-communicable |
| A00-B99 | other infectious diseases | communicable |
| A15-A19 | tb | aids-tb |
| X60-X84 | suicide | external |
| Varies | other injuries | external |
| C53 | cervical cancer | non-communicable |
| I60-I69 | stroke | non-communicable |
| B50-B54 | malaria | communicable |
| J45 | asthma | non-communicable |
| C18-C20 | colorectal cancer | non-communicable |
| X85-Y09 | homicide | external |
| A09 | diarrhea/dysentery | communicable |
| C50 | breast cancer | non-communicable |
| C81-C96 | leukemia/lymphomas | non-communicable |
| X40-X49 | poisonings | external |
| C61 | prostate cancer | non-communicable |
| C15 | esophageal cancer | non-communicable |
| C16 | stomach cancer | non-communicable |
| T63 | bite of venomous animal | external |

## A.2 PPI and PPI++: overview

We formalize our problem setting while reviewing the prediction-powered inference PPI and PPI++ approach (Angelopoulos et al., 2023a;b). The goal of PPI is to estimate a $d$-dimensional parameter $\theta$ with $n$ labeled data points $(X_{li}, Y_{li}) \overset{\text{iid}}{\sim} \mathbb{P}, i \in \{1, \dots, n\}$, and $N$ unlabeled data points $X_{ui} \overset{\text{iid}}{\sim} \mathbb{P}_X, i \in \{1, \dots, N\}$. Additionally, there is a black-box model $f$ that uses the features to predict the outcomes based on labeled and unlabeled data, denoted as $\hat{Y}_l^f = (\hat{Y}_{l1}^f, \dots, \hat{Y}_{ln}^f)$ and $\hat{Y}_u^f = (\hat{Y}_{u1}^f, \dots, \hat{X}_{uN}^f)$. The prediction rule $f$ is assumed to be independent of the observed data. While PPI performs inference on a correction term to incorporate these black-box predictions; PPI++ improves the methodology to be more computationally efficient and develops a modification that adapts to the accuracy of the prediction rule $f$ by introducing an additional parameter $\lambda$ that interpolates between classical and prediction-powered inference.

The starting point of PPI and PPI++ approach is to define the population losses:

$$L(\theta) = \mathbb{E}[l_\theta(X, Y)], \text{ and}$$

$$L^f(\theta) = \mathbb{E}[l_\theta(X, \hat{Y}^f)].$$

It is recognized that

$$L^{f_l}(\theta) = \mathbb{E}[l_\theta(X_l, \hat{Y}_l^f)] = \mathbb{E}[l_\theta(X_u, \hat{Y}_u^f)] = L^{f_u}(\theta).$$

Therefore the "rectified" loss is defined as:

$$L^{PPI}(\theta) = L_n(\theta) + L_N^{f_u}(\theta) - L_n^{f_l}(\theta)$$

where

$$L_n(\theta) = \frac{1}{n} \sum_{i=1}^n l_\theta(X_{li}, Y_{li}),$$

$$L_n^{f_l}(\theta) = \frac{1}{n} \sum_{i=1}^n l_\theta(X_{li}, \hat{Y}_{li}^f)), \text{ and}$$

$$L_N^{f_u}(\theta) = \frac{1}{N} \sum_{i=1}^N l_\theta(X_{ui}, \hat{Y}_{ui}^f).$$

The rectified loss leads to the prediction-powered point estimate:

$$\hat{\theta}^{PPI} = \arg\min_{\theta \in \mathbb{R}^d} L^{PPI}(\theta).$$

PPI constructs the confidence set $C_\alpha^{PPI} = \{\theta \in \mathbb{R}^d \text{ s.t. } \mathbf{0} \in C_\alpha^\nabla(\theta)\}$ for true parameter $\theta^*$ by forming $(1 - \alpha)$-confidence sets, $C_\alpha^\nabla(\theta)$, for the gradient of the loss $\nabla L^{PPI}(\theta)$. This approach works well for a fixed $\theta$ but suffers from computational issue when forming the confidence set for every $\theta \in \mathbb{R}^d$. PPI++ derives the asymptotic normality about $\hat{\theta}$. This result allows the construction of $(1 - \alpha)$-confidence intervals $C_\alpha^{PPI} = \{\hat{\theta}_j^{PPI} \pm z_{1-\alpha/2} \times \hat{\sigma}_j / \sqrt{n}\}$, where $\hat{\sigma}_j^2$ is a consistent estimate of the asymptotic variance of $\hat{\theta}_j^{PPI}$ and $z_q$ is the $q$-quantile of standard normal distribution. Similar confidence intervals centered around $\hat{\theta}^{PPI}$ can be constructed when the inference is performed on the whole $\theta^*$ vector rather than a single coordinate $\theta_j^*$, which results in far more efficient algorithm.

Moreover, PPI++ generalizes PPI to allow the inference to adapt to the accuracy of the supplied predictions, yielding estimations that are never worse than the classical inference. This approach is called "power tuning" by incorporating a tuning parameter $\lambda \in [0, 1]$ in the rectified loss, leading to the following estimator:

$$\hat{\theta}_\lambda^{\texttt{PPI++}} = \arg\min_\theta L_\lambda^{\texttt{PPI++}}(\theta),$$

where

$$L_\lambda^{\texttt{PPI++}}(\theta) = L_n(\theta) + \lambda(L_N^{f_u}(\theta) - L_n^{f_l}(\theta)).$$

PPI++ recovers the original PPI approach when $\lambda = 1$ and reduces to classical inference when $\lambda = 0$. One can adaptively choose a data-dependent tuning parameter $\hat{\lambda}$ to maximize the estimation and inferential efficiency.

### A.3 `multiPPI++`

In $K$-class multinomial classification problem with outcomes $Y_i \in \{0, \ldots, K-1\}$ and covariates $X_i \in \mathbb{R}^d$, the parameter of interests $\theta$ is a $d(K-1)$-dimensional vector whose $k$-th block $\theta_k \in \mathbb{R}^d$ of components represent parameters for class $k \in \{1, \ldots, K-1\}$ excluding the reference class to avoid overparameterization. The loss function in classical inference takes the form

$$l_\theta(X, Y) = -\frac{1}{n} \sum_{i=1}^n X_i^T \theta_{Y_i} + \log\left(\sum_{k=1}^{K-1} e^{X_i^T \theta_k}\right),$$

---

**Algorithm 1:** `multiPPI++`: Prediction-powered inference for Multinomial Classification

---

**Input:** labeled $K$-category COD data $\{(X_{li}, Y_{li})\}_{i=1}^{n}$, unlabeled data $\{X_{ui}\}_{i=1}^{N}$, NLP
model $f$, significance level $\alpha \in (0, 1)$, coefficient index $j \in [d(K - 1)]$

1. Optimally select tuning parameter $\hat{\lambda}$   // set tuning parameter

2. $\hat{\theta}_{\hat{\lambda}}^{\mathfrak{m}} = \underset{\theta \in \mathbb{R}^{d(K-1)}}{\arg \min} L_{\hat{\lambda}}^{\mathfrak{m}}(\theta)$   // multiPPI++ estimator

3. $\hat{H} = \frac{1}{N+n}(\sum_{i=1}^{n} \psi''(X_{li}^{T}\hat{\theta}_{\hat{\lambda}}^{\mathfrak{m}})X_{li}X_{li}^{T} + \sum_{i=1}^{N} \psi''(X_{ui}^{T}\hat{\theta}_{\hat{\lambda}}^{\mathfrak{m}})X_{ui}X_{ui}^{T})$, where
   $\psi(\theta, x) = \log\left(\sum_{k=1}^{K-1} e^{x^{T}\theta_{k}}\right), \theta_{k} \in \mathbb{R}^{d}$   // empirical Hessian

4. $\hat{\Sigma} = \hat{H}^{-1}(\frac{n}{N}\hat{V}_{f} + \hat{V}_{\Delta})\hat{H}^{-1}$, where

   $\hat{V}_{f} = \hat{\lambda}^{2}\widehat{\text{Cov}}_{N+n}((\psi'(X_{li}^{T}\hat{\theta}_{\hat{\lambda}}^{\mathfrak{m}}) - \hat{Y}_{li}^{f}))X_{li})$ and
   $\hat{V}_{\Delta} = \widehat{\text{Cov}}_{n}((1 - \hat{\lambda})(\psi'(X_{li}^{T}\hat{\theta}_{\hat{\lambda}}^{\mathfrak{m}}) + (\hat{\lambda}\hat{Y}_{li}^{f} - Y_{li})X_{li})$   // covariance estimator

**Output:**
Prediction-powered point estimates $\hat{\theta}_{\hat{\lambda}}^{\mathfrak{m}}$ and confidence interval
$\mathcal{C}_{\alpha}^{\mathfrak{m}} = \left(\hat{\theta}_{\hat{\lambda}, j}^{\mathfrak{m}} \pm z_{1-\alpha/2}\sqrt{\hat{\Sigma}_{jj}/n}\right)$ for coordinate $j$

---

Defining

$$\psi(\theta, x) = \log\left(\sum_{k=1}^{K-1} e^{x^{T}\theta_{k}}\right),$$

and the `multiPPI++` loss is given by

$$\begin{aligned}
L_{\lambda}^{\mathfrak{m}}(\theta) &= L_{n}(\theta) + \lambda(L_{N}^{f_{u}}(\theta) - L_{n}^{f_{l}}(\theta)) \\
&= -\frac{1}{n}\sum_{i=1}^{n}(X_{li}\theta_{Y_{li}} - \psi(\theta, X_{li})) \\
&\quad - \frac{\lambda}{N}\sum_{i=1}^{N}(X_{ui}^{T}\theta_{\hat{Y}_{ui}^{f}} - \psi(\theta, X_{ui})) + \frac{\lambda}{n}\sum_{i=1}^{n}(X_{li}^{T}\theta_{\hat{Y}_{li}^{f}} - \psi(\theta, X_{li}))
\end{aligned}$$

Angelopoulos et al. (2023b) shows that the parameter $\lambda$ can be optimally tuned to minimize
the asymptotic variance of the prediction-powered estimate, leading to better estimation
and inference than both the classical and `PPI` strategies. Particularly, in finite samples, we
can obtain the plug-in estimate

$$\hat{\lambda} = \frac{1}{2(1 + \frac{n}{N})} \times \frac{\text{Tr}\left(\hat{H}_{\hat{\theta}_{PPI}}^{-1}\left(\widehat{\text{Cov}}_{n}(\nabla l_{\hat{\theta}_{PPI}}, \nabla l_{\hat{\theta}_{PPI}}^{f}) + \widehat{\text{Cov}}_{n}(\nabla l_{\hat{\theta}_{PPI}}^{f}, \nabla l_{\hat{\theta}_{PPI}})\right)\hat{H}_{\hat{\theta}_{PPI}}^{-1}\right)}{\text{Tr}\left(\hat{H}_{\hat{\theta}_{PPI}}^{-1}\widehat{\text{Cov}}_{N+n}(\nabla l_{\hat{\theta}_{PPI}}^{f})\hat{H}_{\hat{\theta}_{PPI}}^{-1}\right)}$$

where $\hat{\theta}_{PPI} = \hat{\theta}_{\lambda}^{\text{PPI++}}$ is obtained by taking a fixed $\lambda \in [0, 1]$.

We summarize the `multiPPI++` adjusted inference procedures for multinomial logistic re-
gression coefficients on predicted CODs by an NLP model $f$ against covariate $X$, under
significance level $\alpha \in (0, 1)$ in Algorithm 1.

### A.4 Parameter Estimates Across All Sites: Full Data 80/20 Split

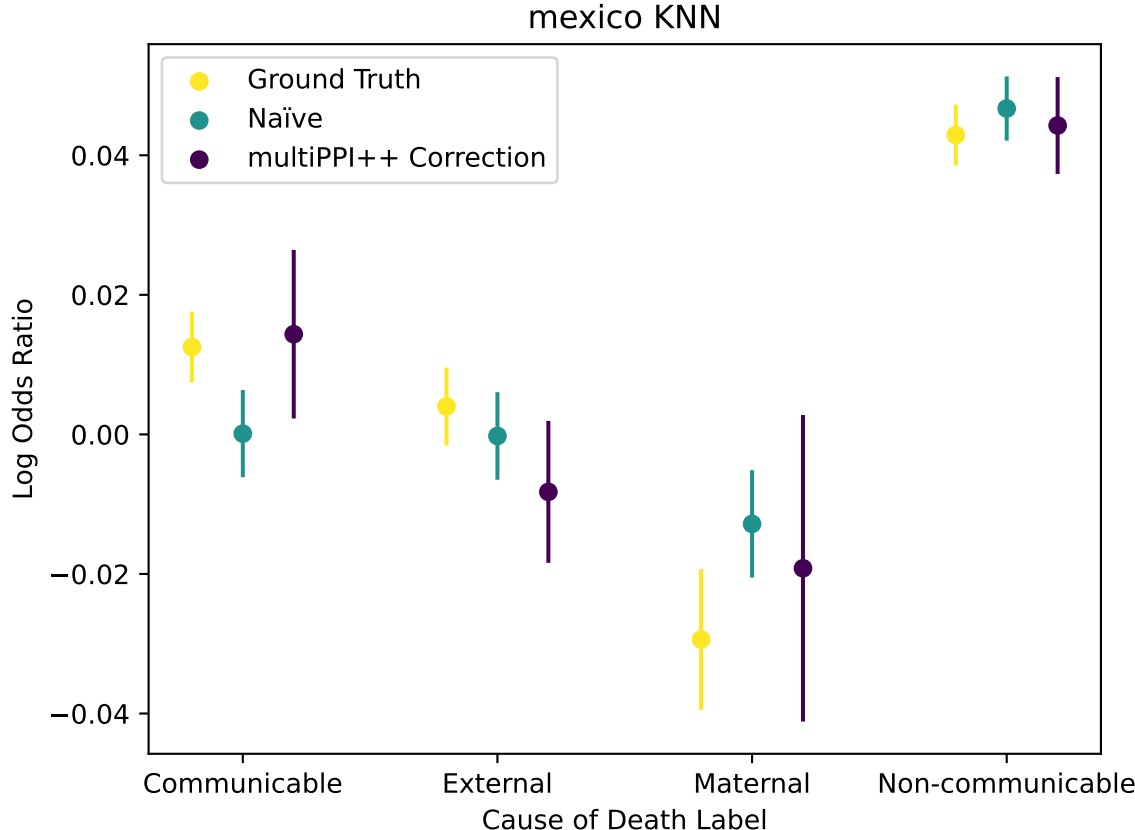

Figure 8: Full Data 80/20 Split: Site/Model 1

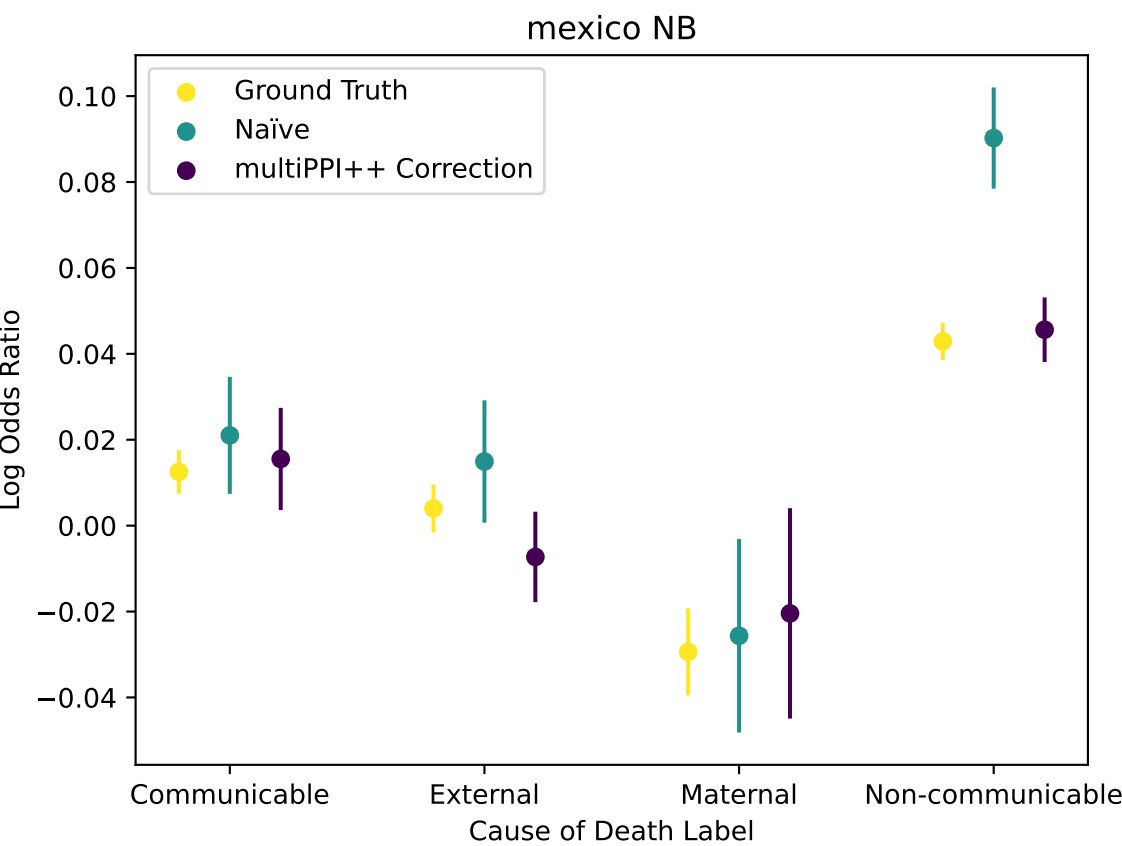

Figure 9: Full Data 80/20 Split: Site/Model 2

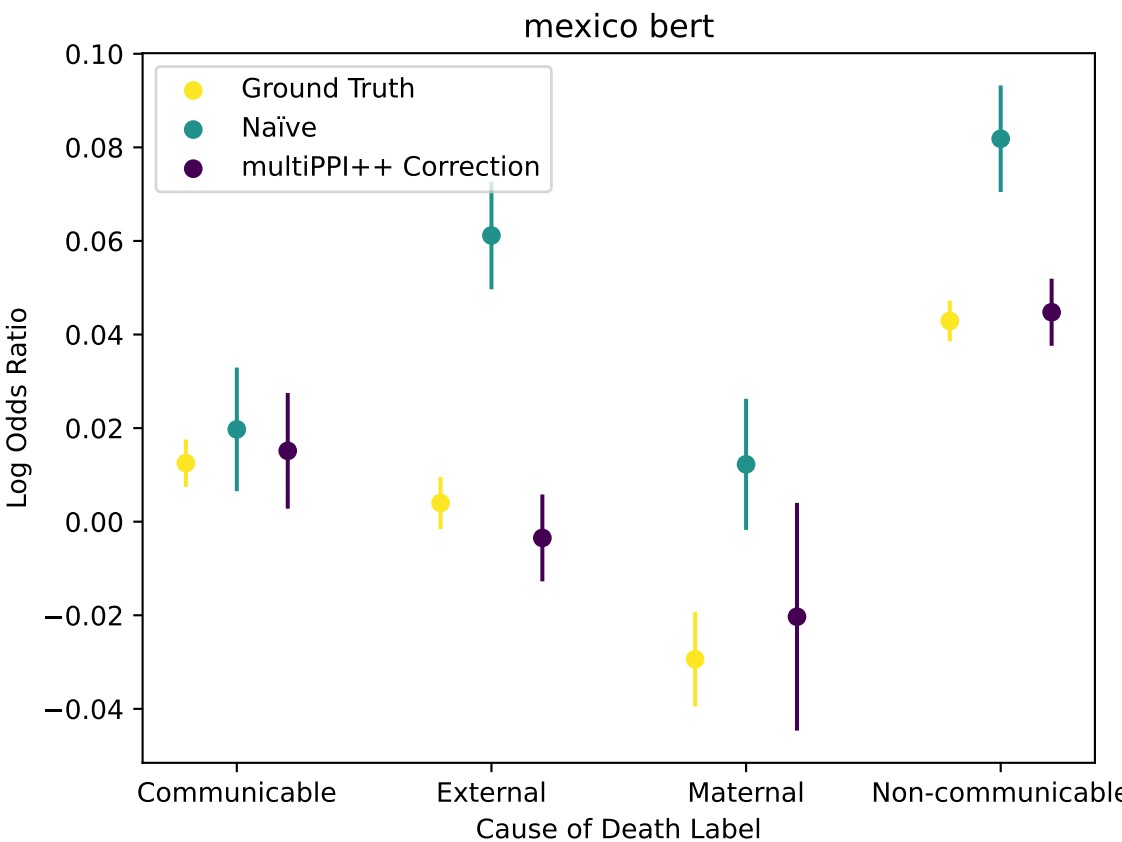

Figure 10: Full Data 80/20 Split: Site/Model 3

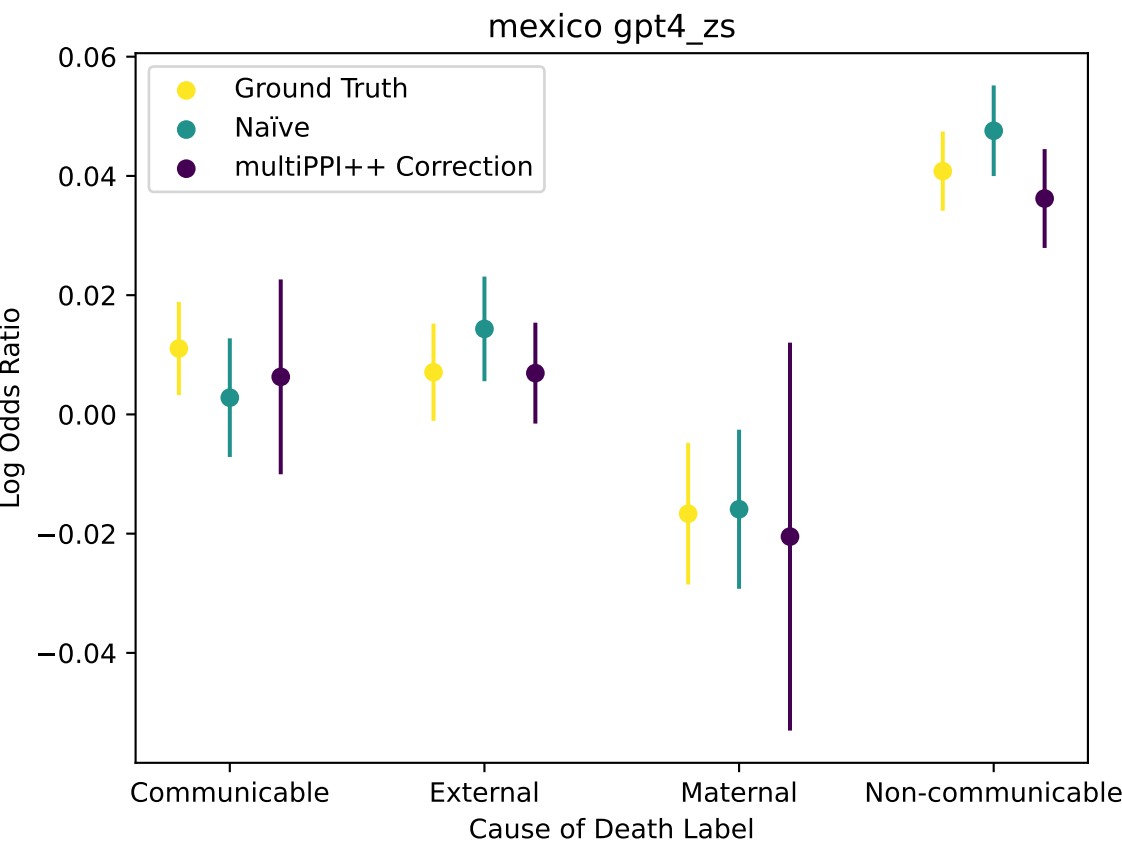

Figure 11: Full Data 80/20 Split: Site/Model 4

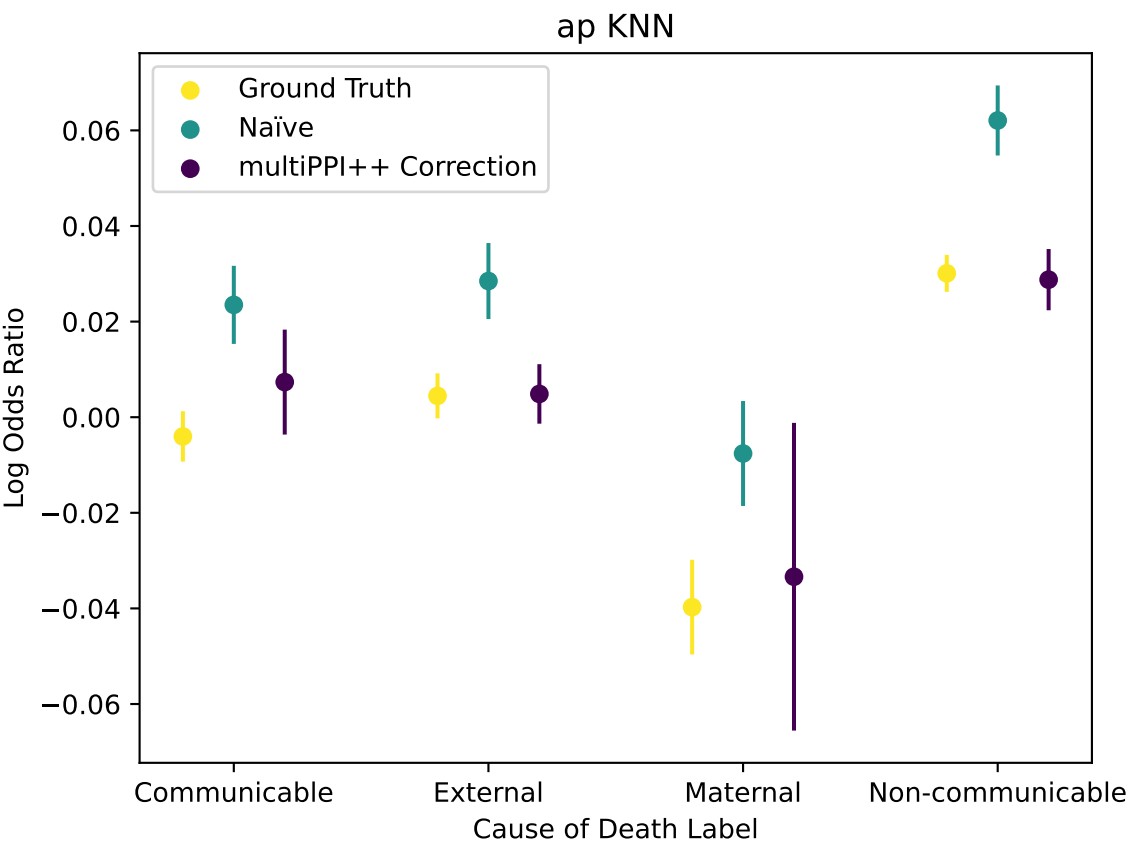

Figure 12: Full Data 80/20 Split: Site/Model 5

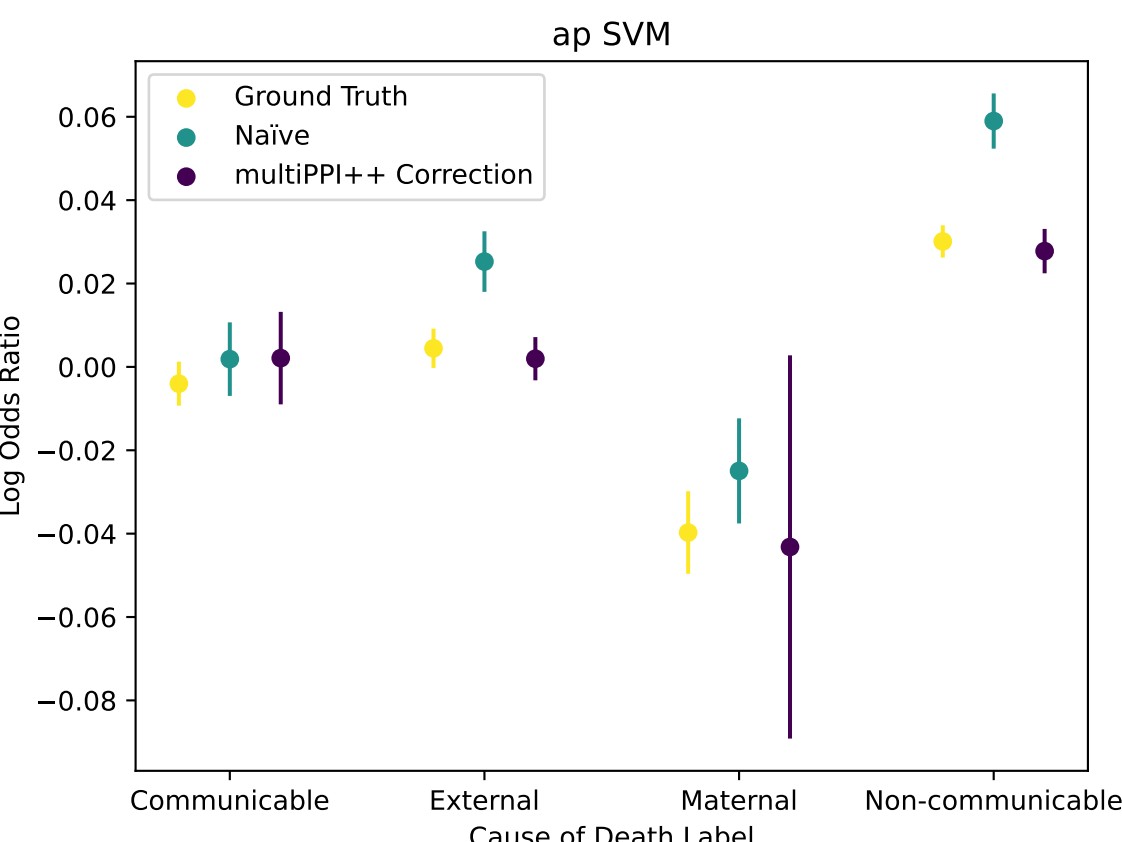

Figure 13: Full Data 80/20 Split: Site/Model 6

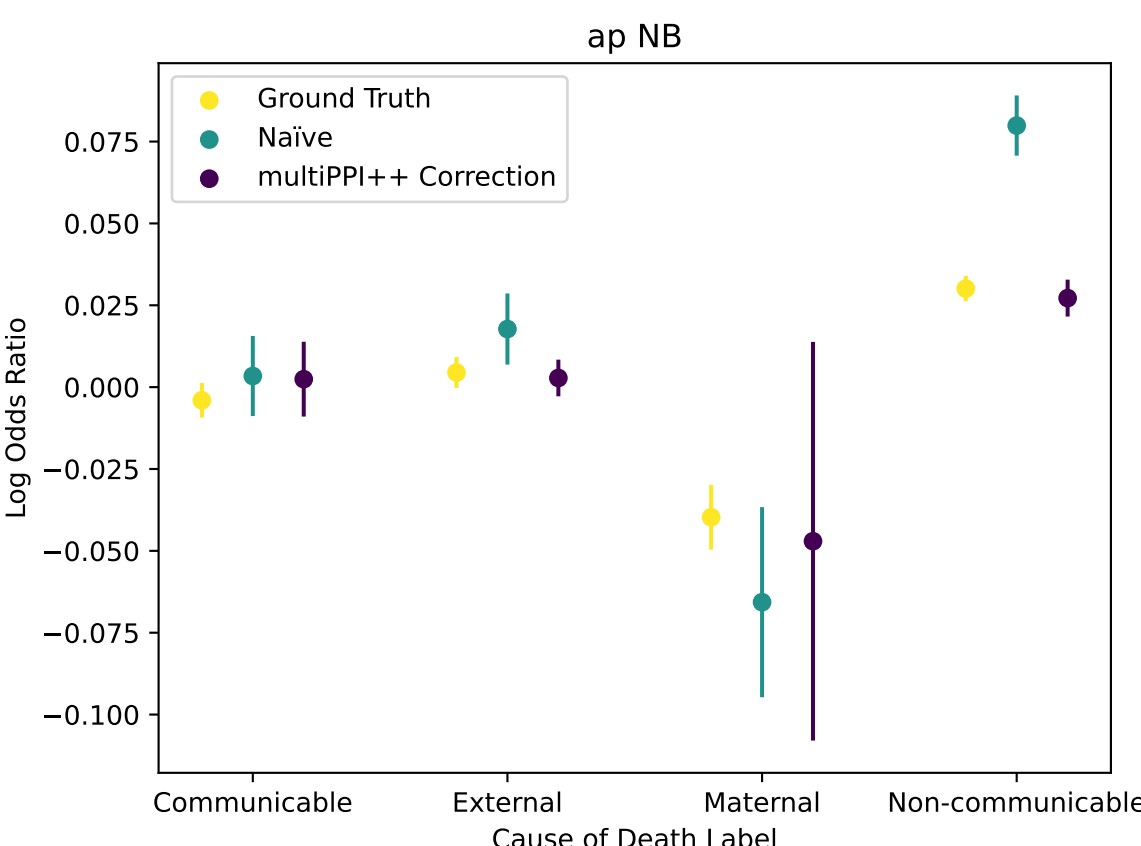

Figure 14: Full Data 80/20 Split: Site/Model 7

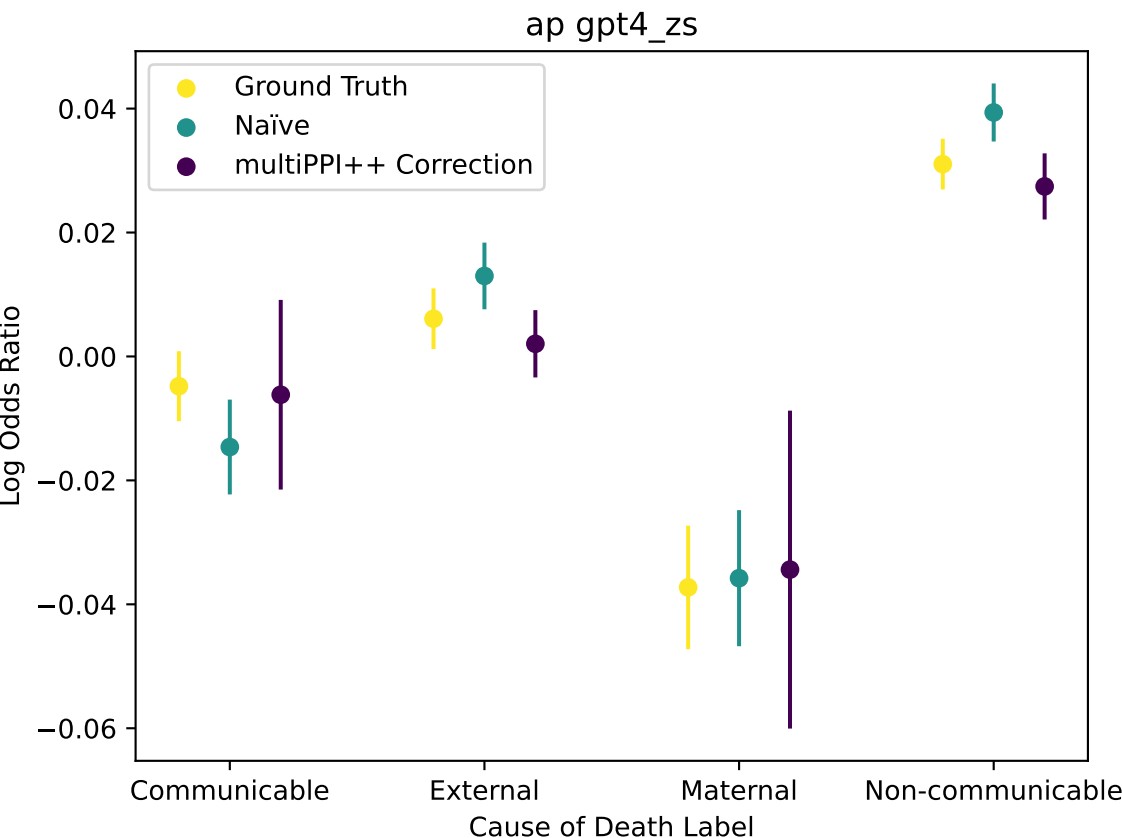

Figure 15: Full Data 80/20 Split: Site/Model 8

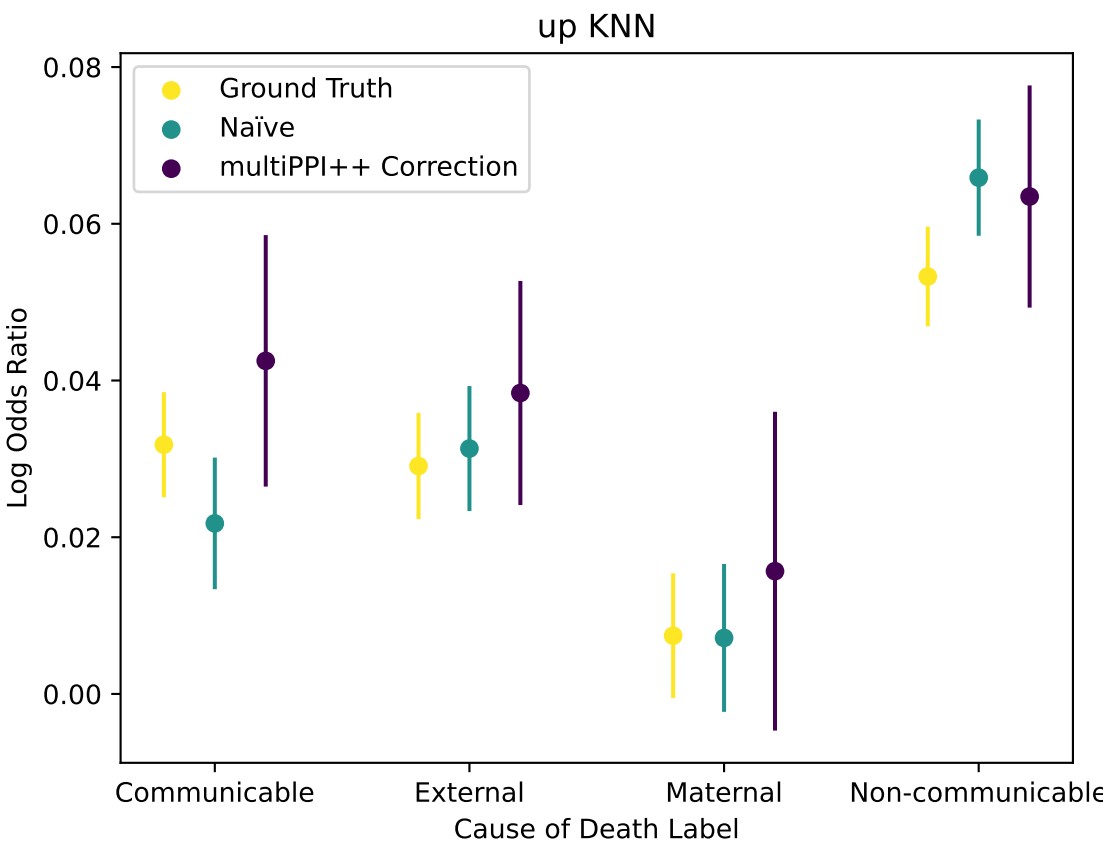

Figure 16: Full Data 80/20 Split: Site/Model 9

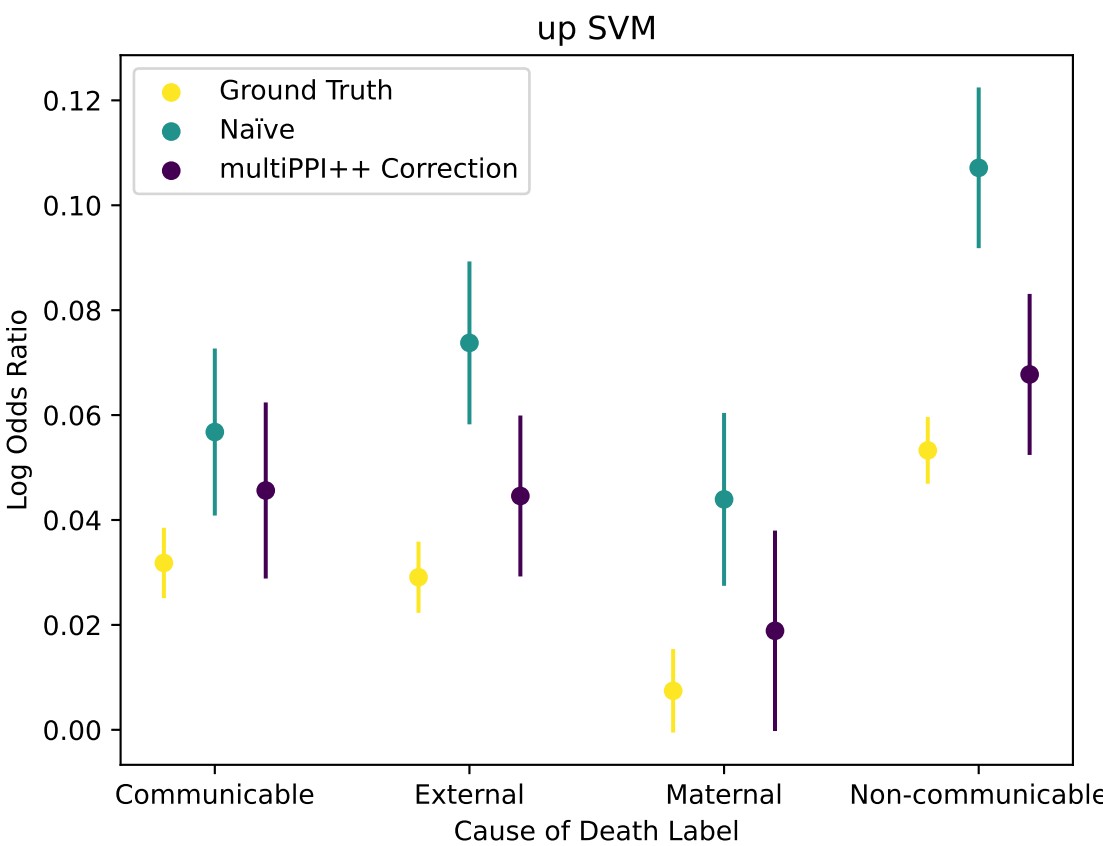

Figure 17: Full Data 80/20 Split: Site/Model 10

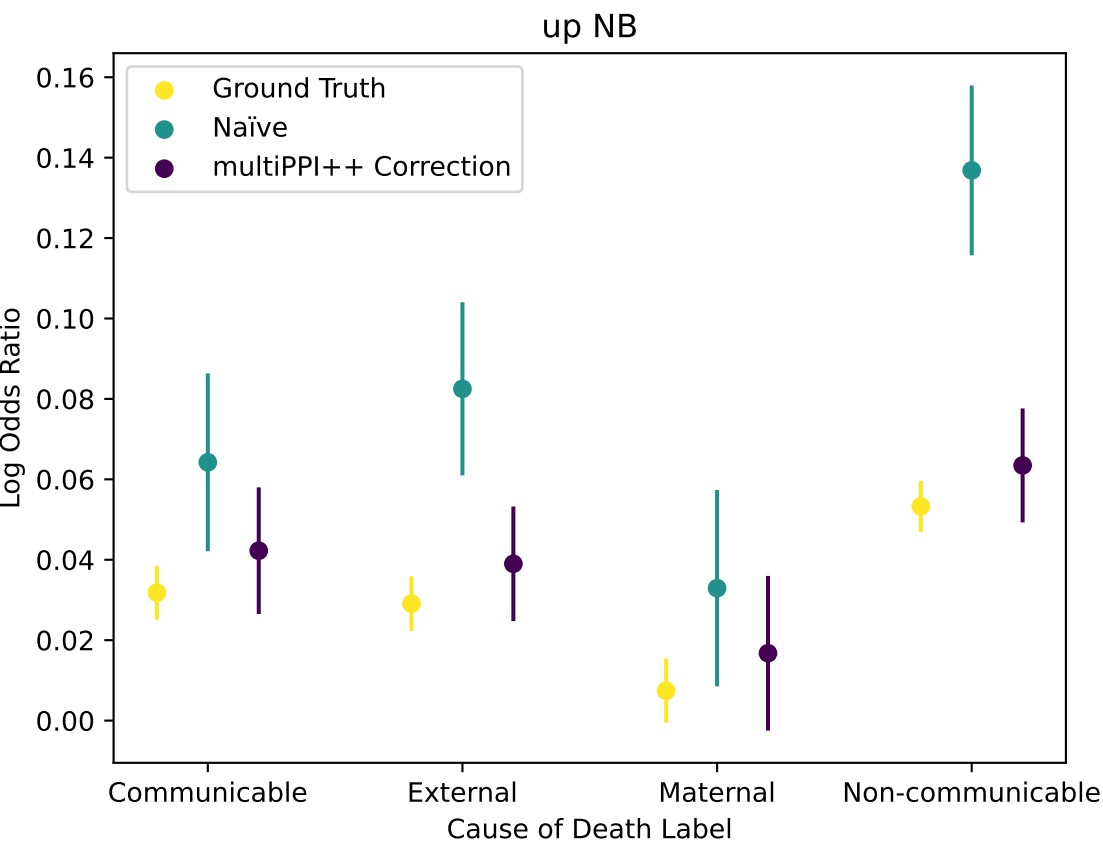

Figure 18: Full Data 80/20 Split: Site/Model 11

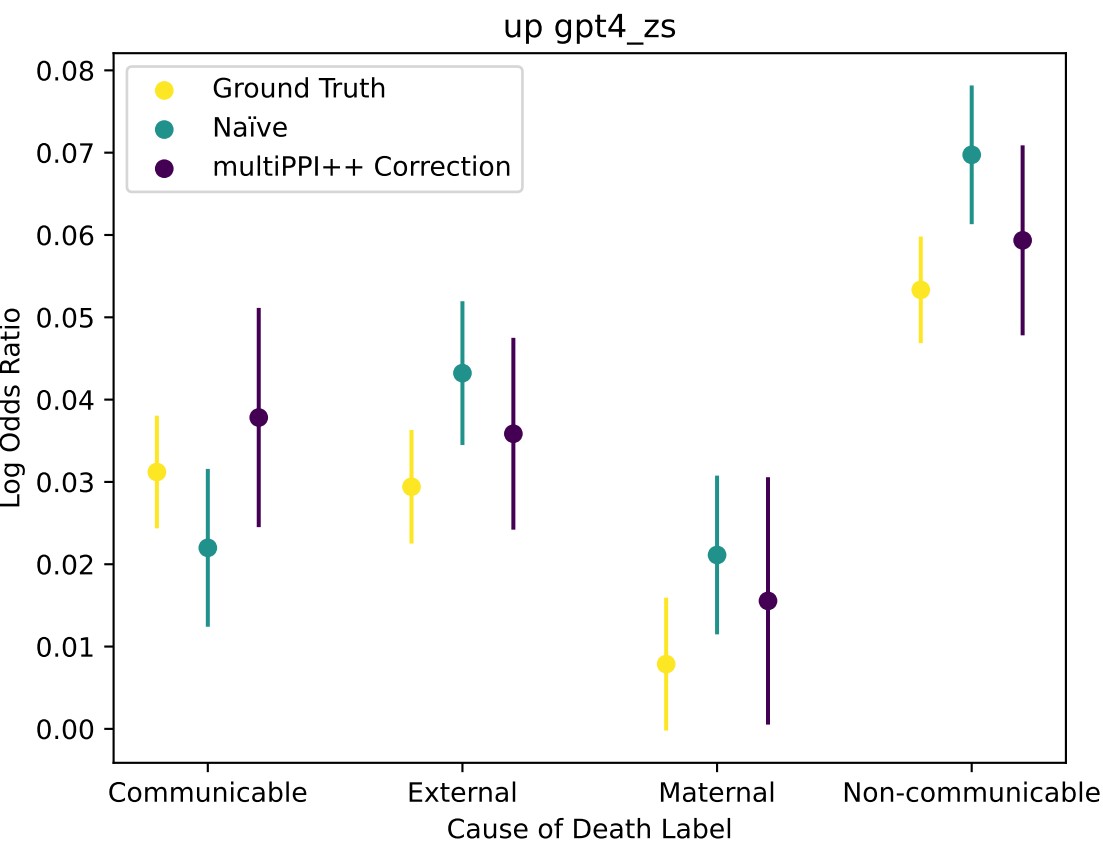

Figure 19: Full Data 80/20 Split: Site/Model 12

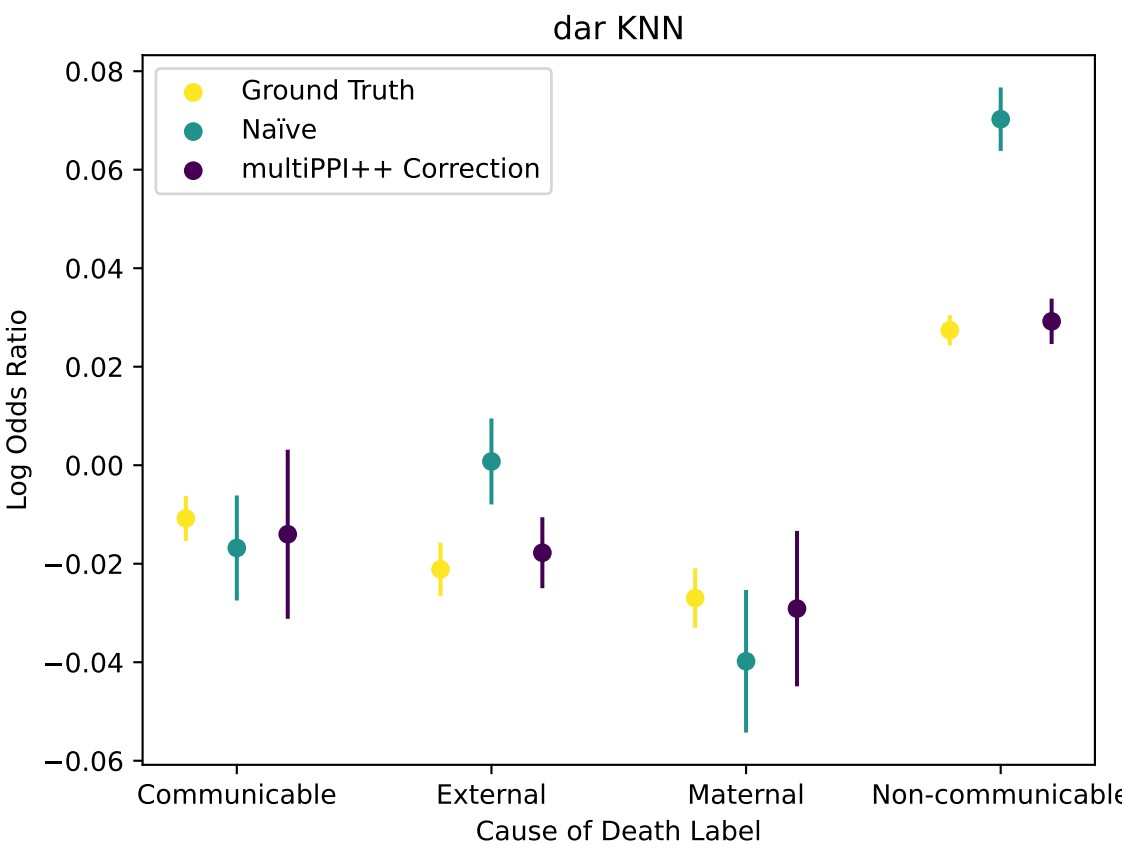

Figure 20: Full Data 80/20 Split: Site/Model 13

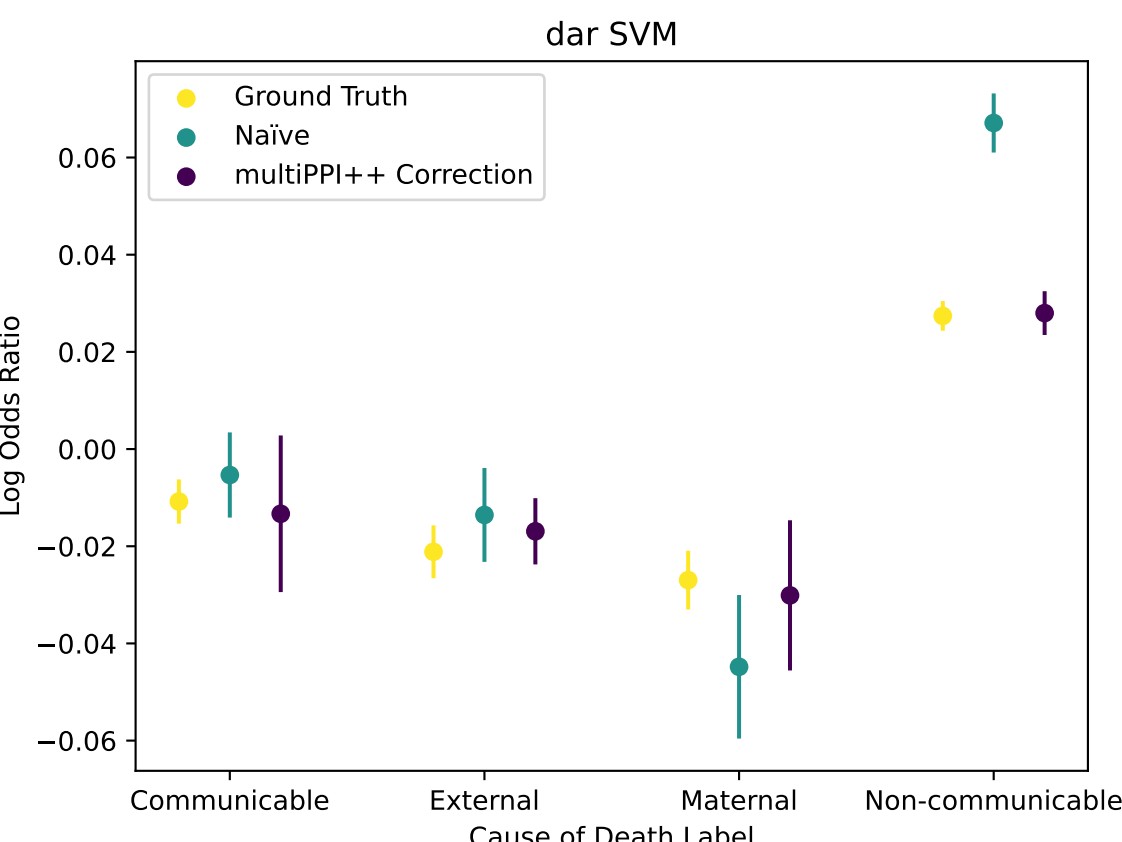

Figure 21: Full Data 80/20 Split: Site/Model 14

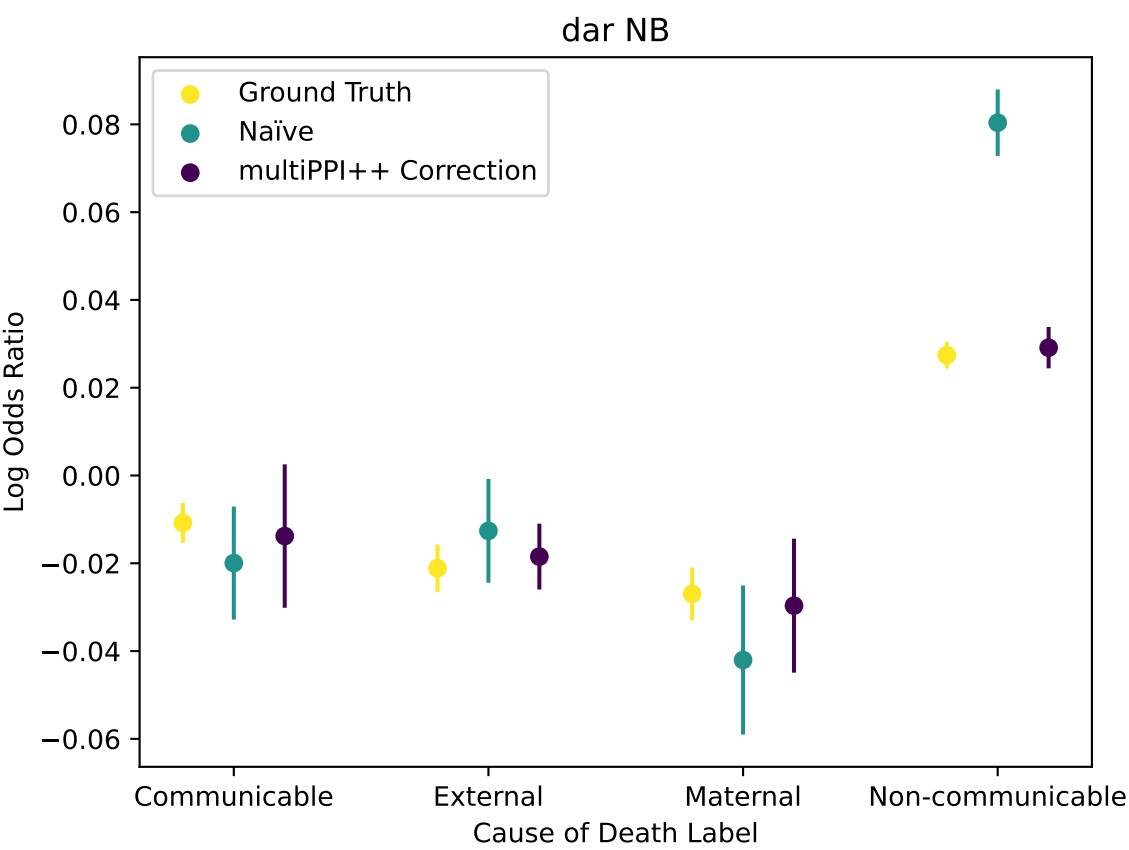

Figure 22: Full Data 80/20 Split: Site/Model 15

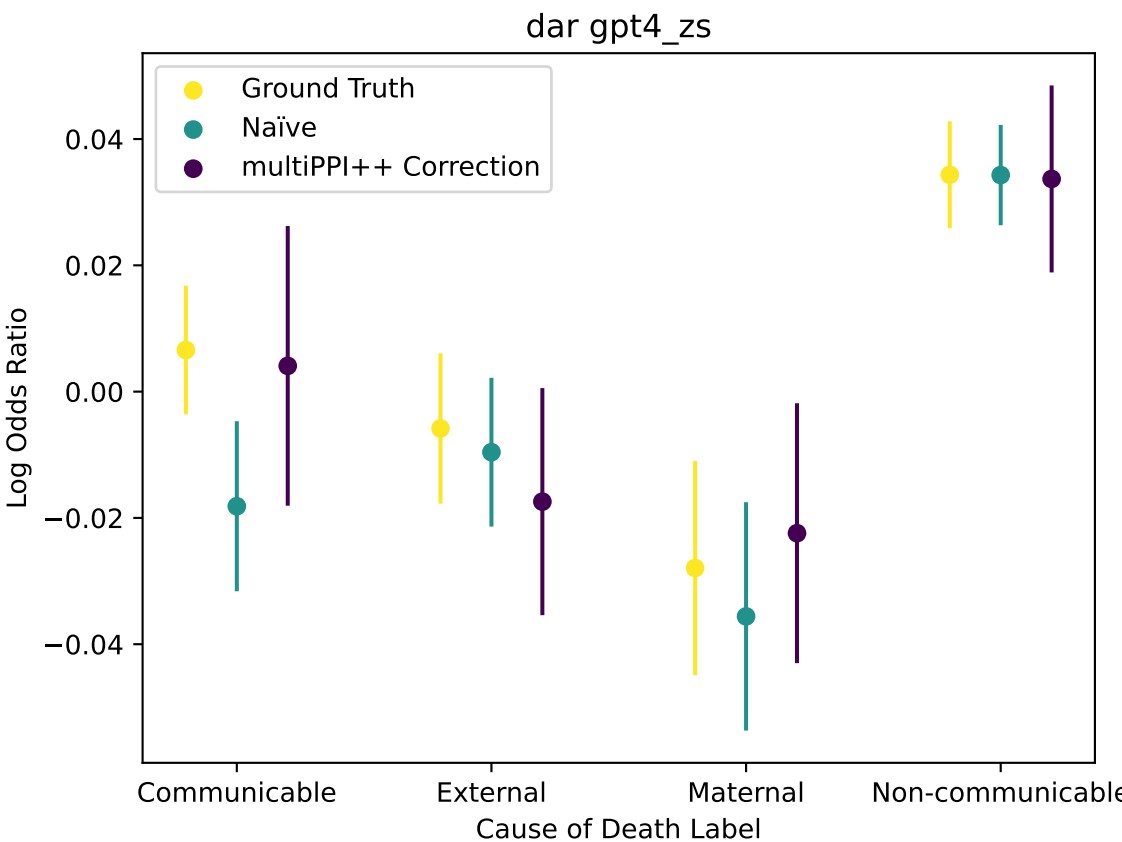

Figure 23: Full Data 80/20 Split: Site/Model 16

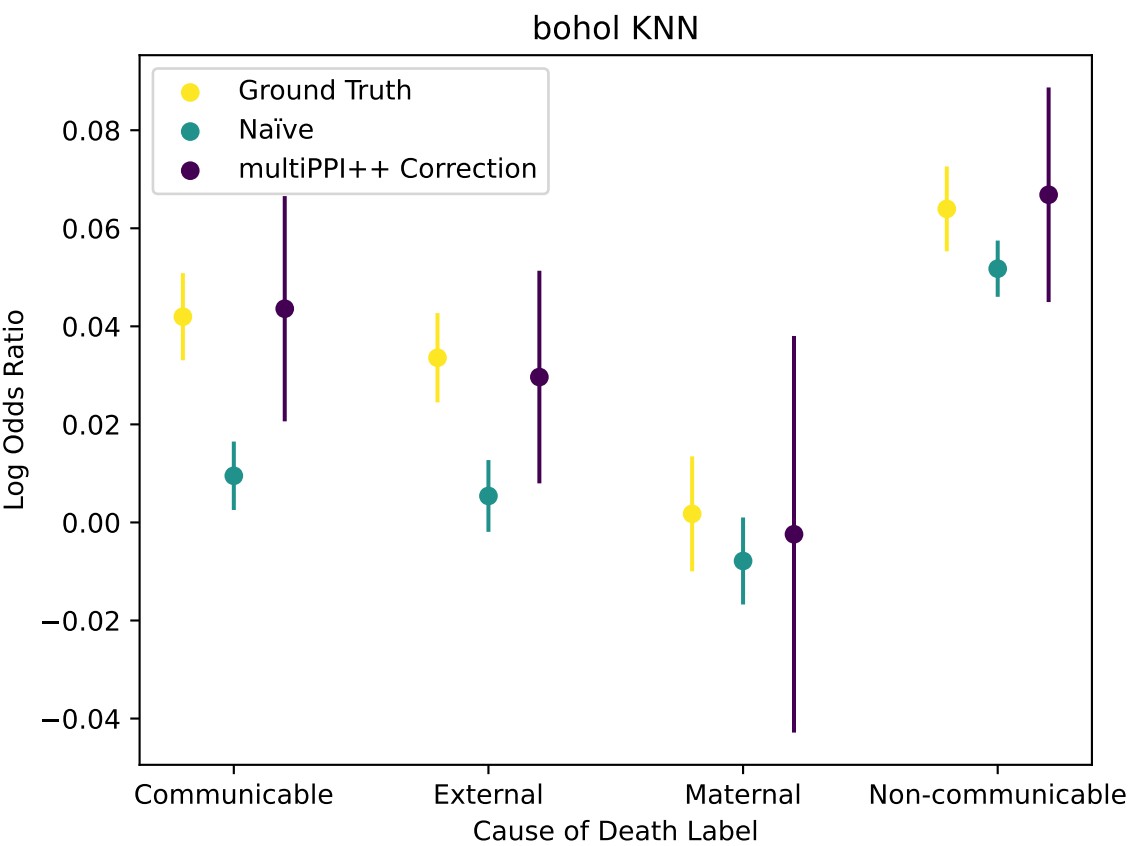

Figure 24: Full Data 80/20 Split: Site/Model 17

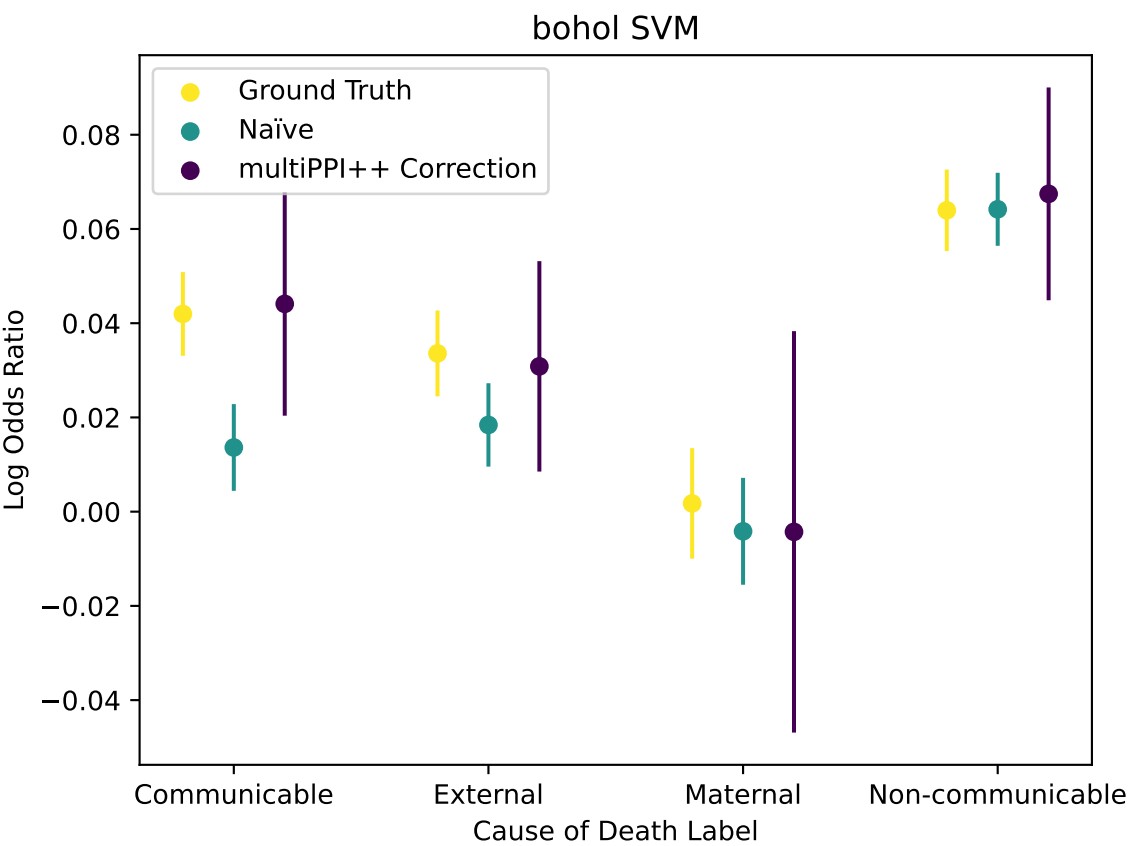

Figure 25: Full Data 80/20 Split: Site/Model 18

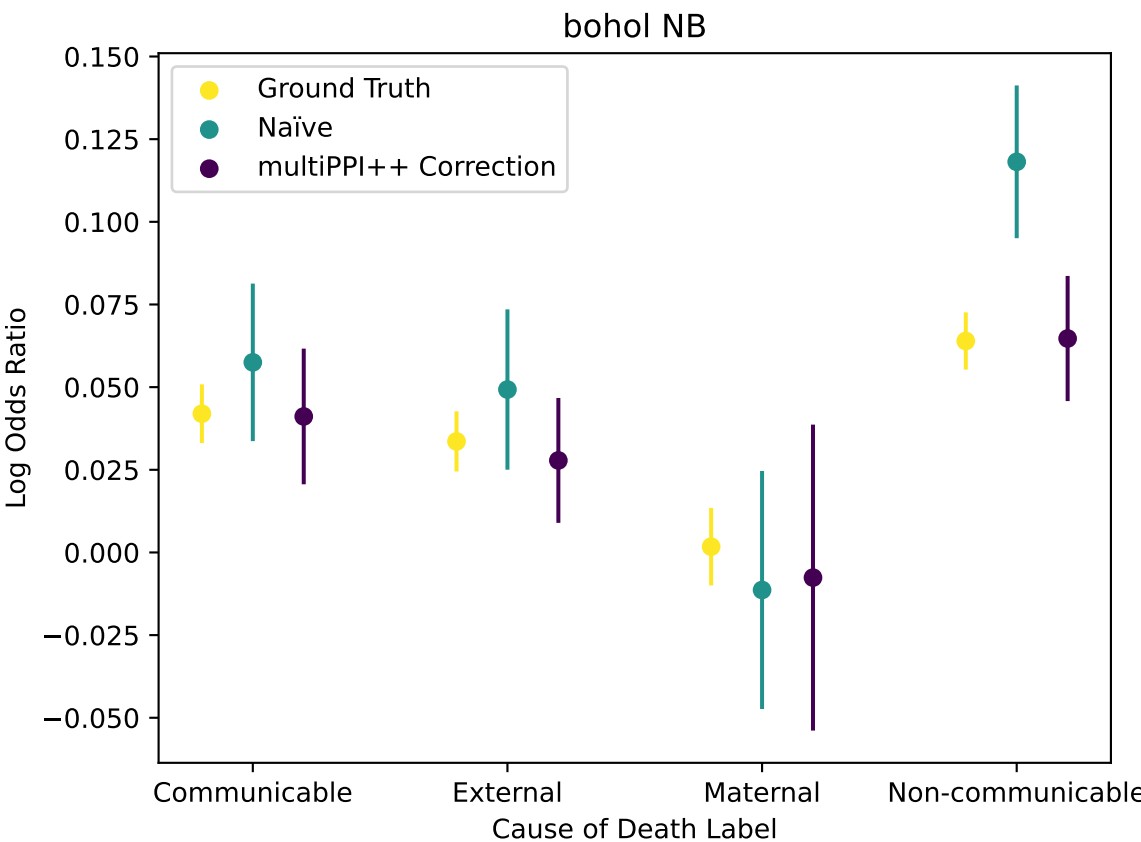

Figure 26: Full Data 80/20 Split: Site/Model 19

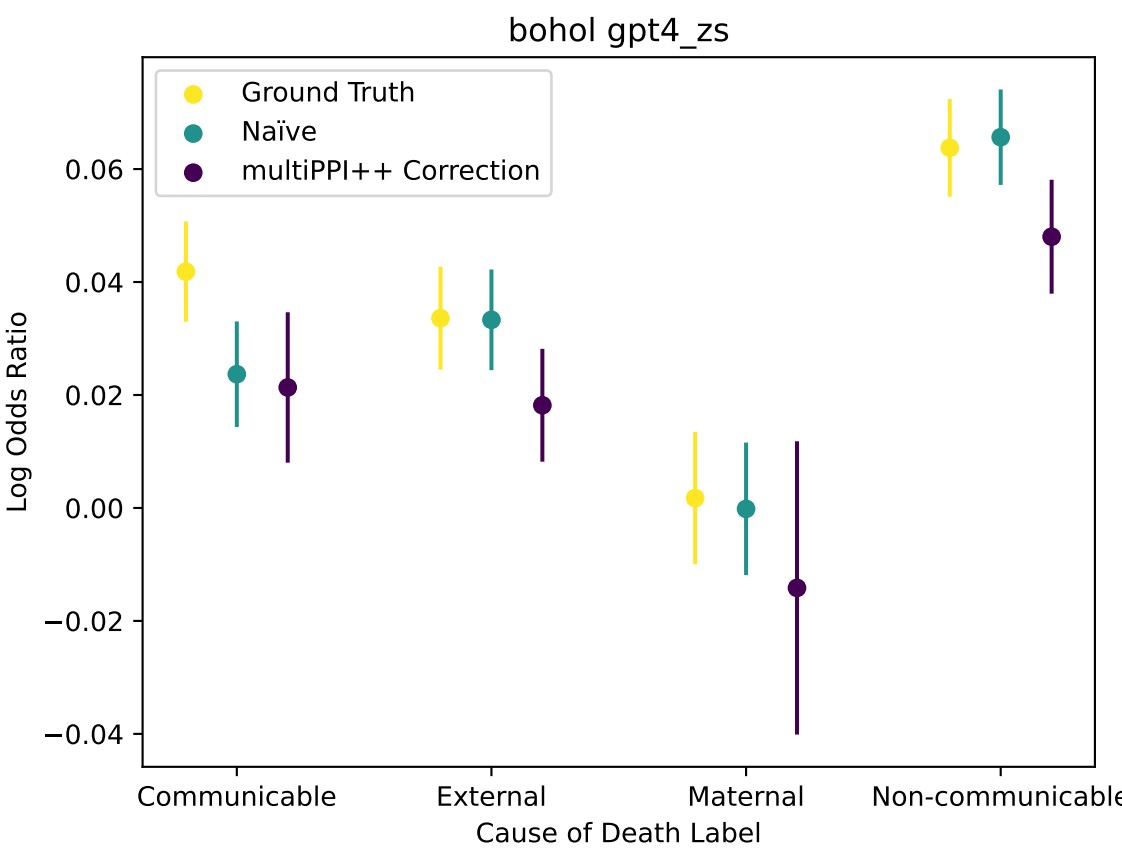

Figure 27: Full Data 80/20 Split: Site/Model 20

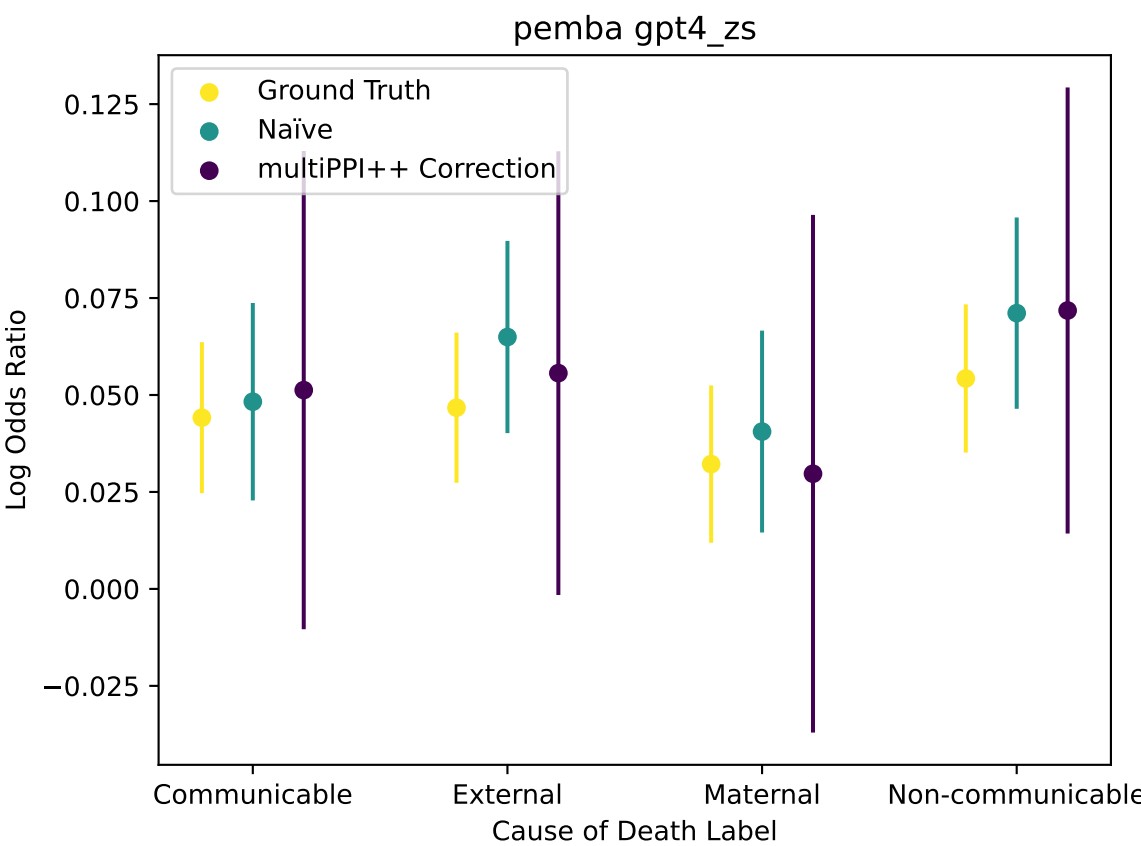

Figure 28: Full Data 80/20 Split: Site/Model 21

### A.5 Sensitivity Analysis

In a sensitivity analysis, we varied our data splitting strategy, which revealed nuanced insights regarding the behavior of the classifier. Notably, when maintaining the split proportions (80/20) within each COD, the `PPI++` classifier exhibited minimal utilization of the labeled data, as indicated by the relatively small values of $\lambda$. Conversely, when employing an 80/20 split on the entire dataset, this lead to a significant information loss for the minority classes, resulting in larger $\lambda$ values for certain splits. Since an (80/20) on the whole dataset more closely resembles what one would see in reality on a prospective study while an (80/20) split by COD is resembles a retrospective study, this illustrates the importance of purposefully choosing an appropriate data splitting strategy to match the desired analysis.

**A.6   Parameter Estimates Across All Sites: Stratified 80/20 Split**

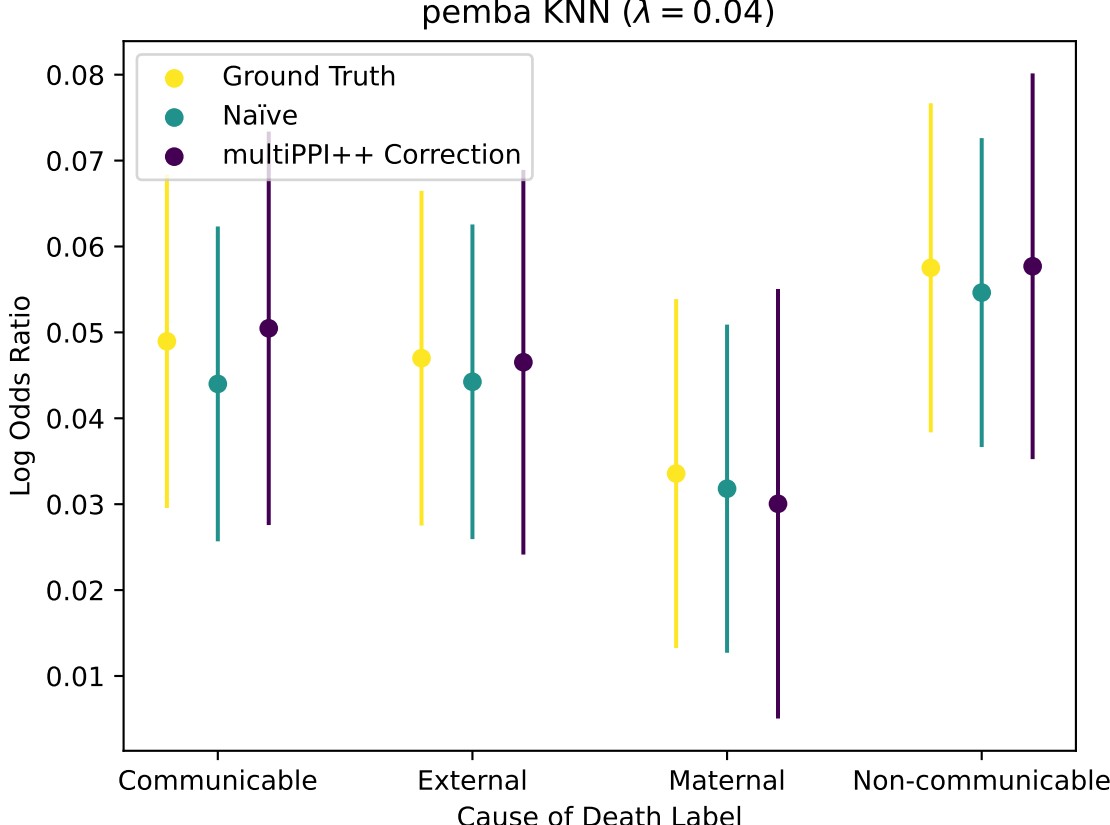

Figure 29: Stratified 80/20 Split: Site/Model 22

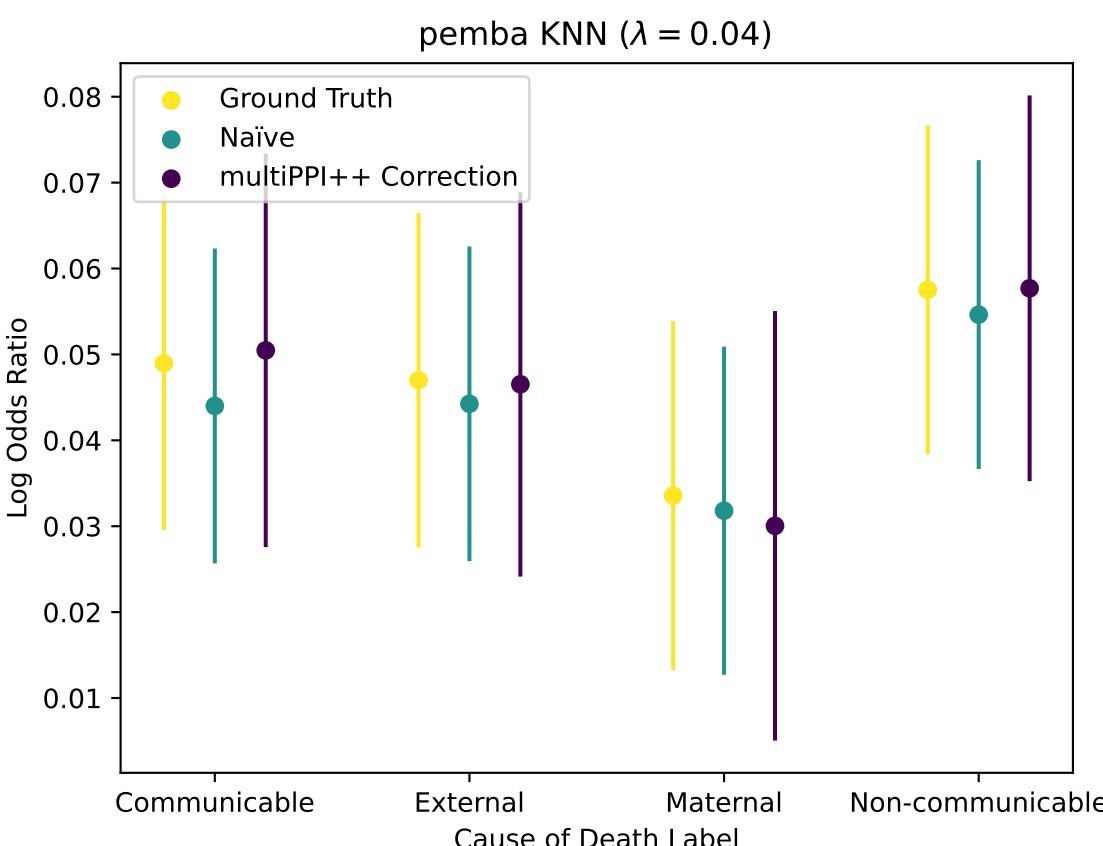

Figure 30: Stratified 80/20 Split: Site/Model 22

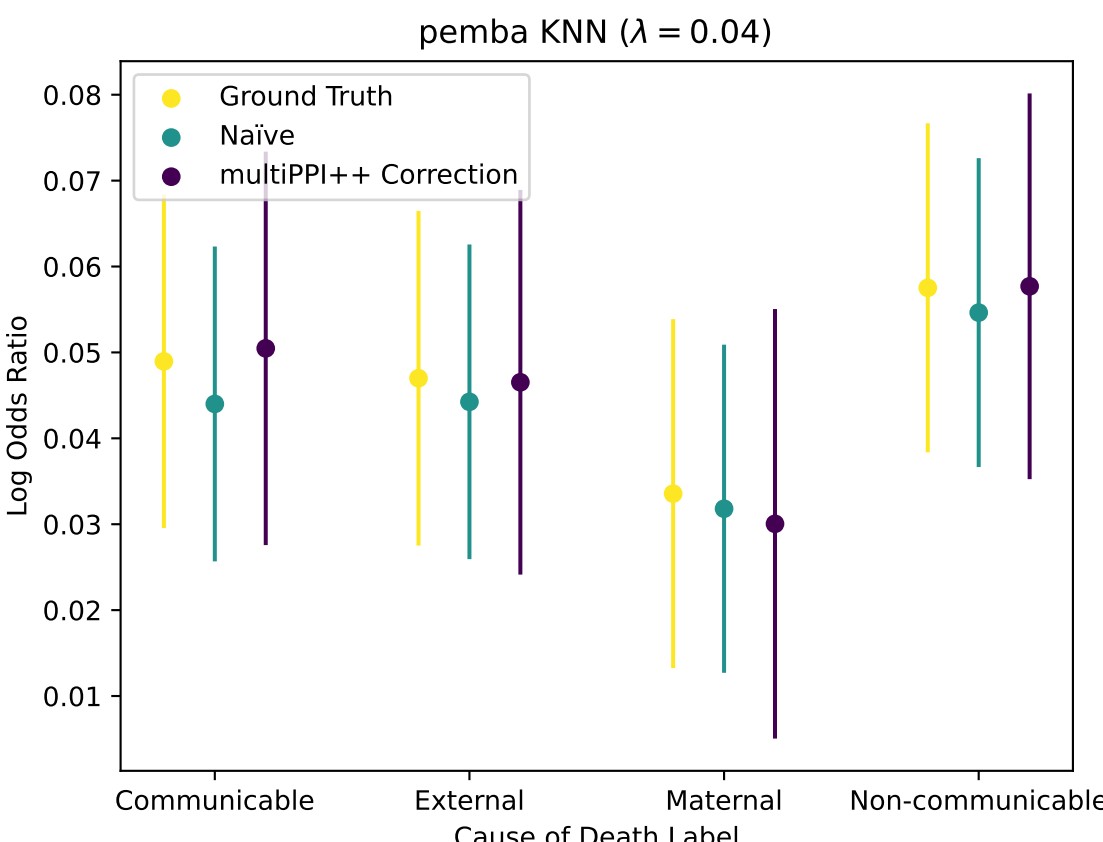

Figure 31: Stratified 80/20 Split: Site/Model 22

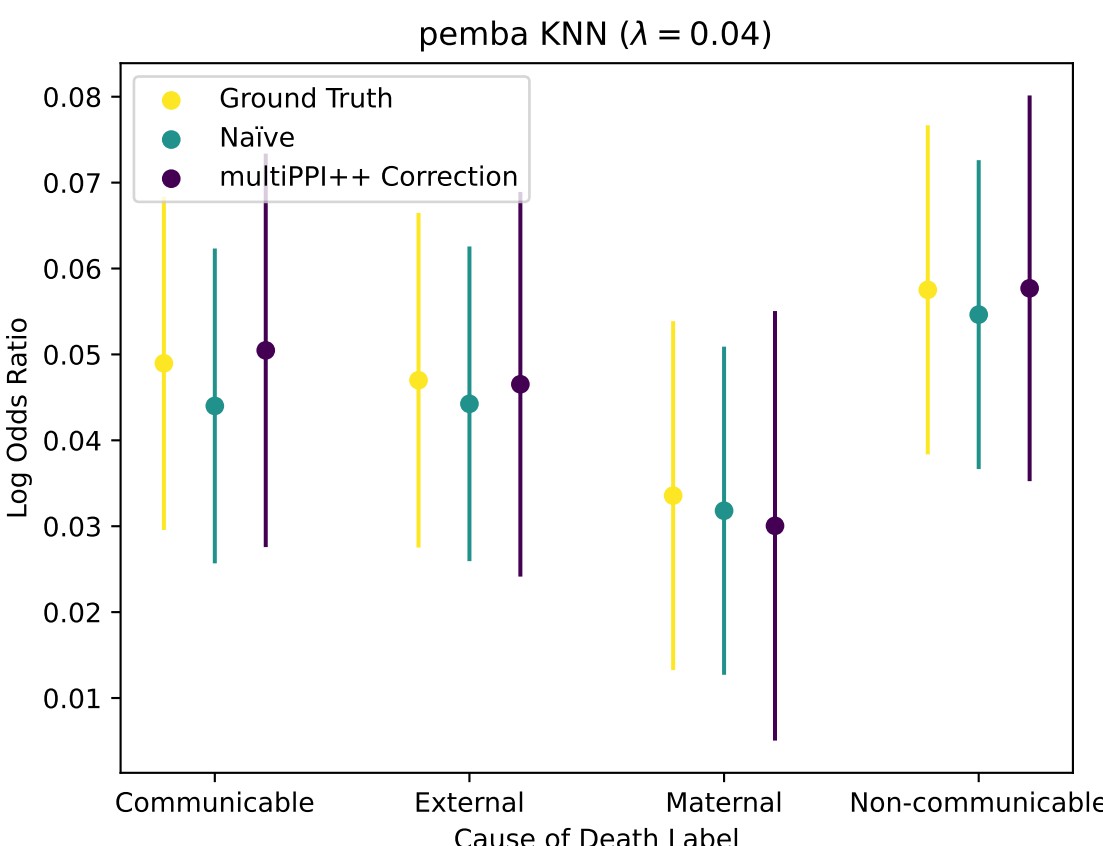

Figure 32: Stratified 80/20 Split: Site/Model 22

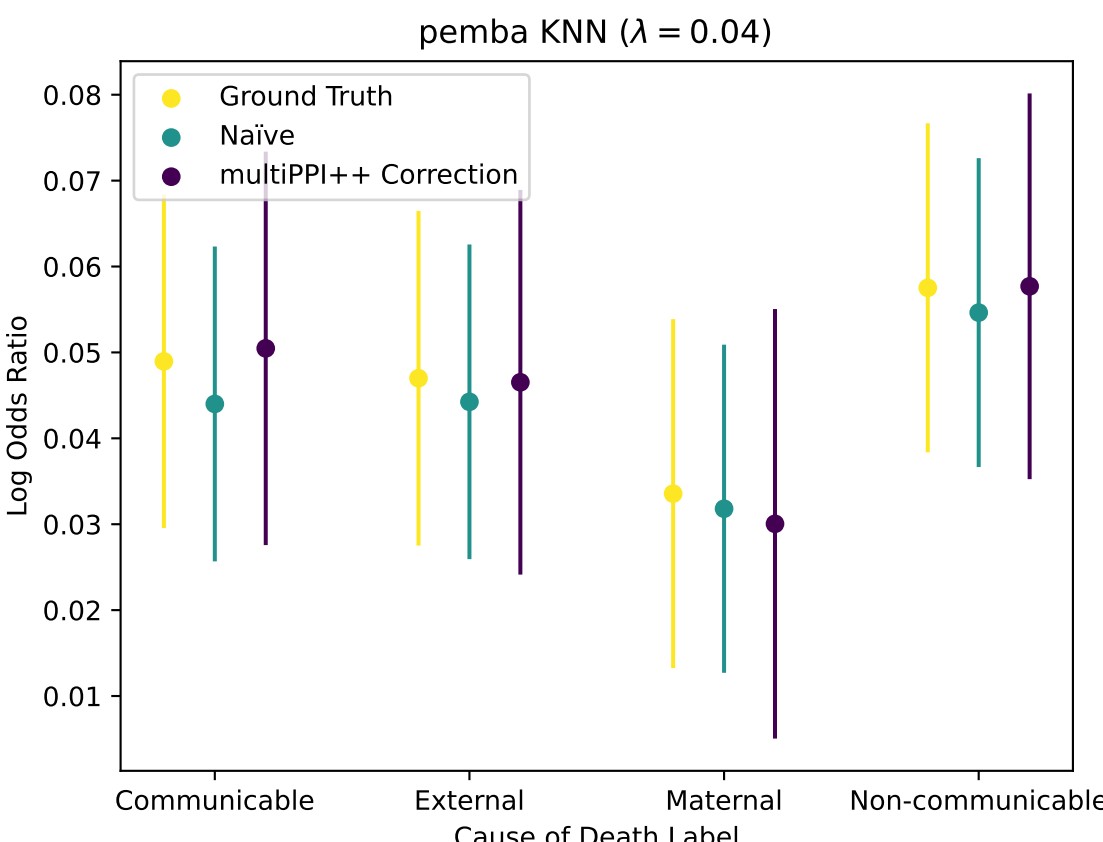

Figure 33: Stratified 80/20 Split: Site/Model 22

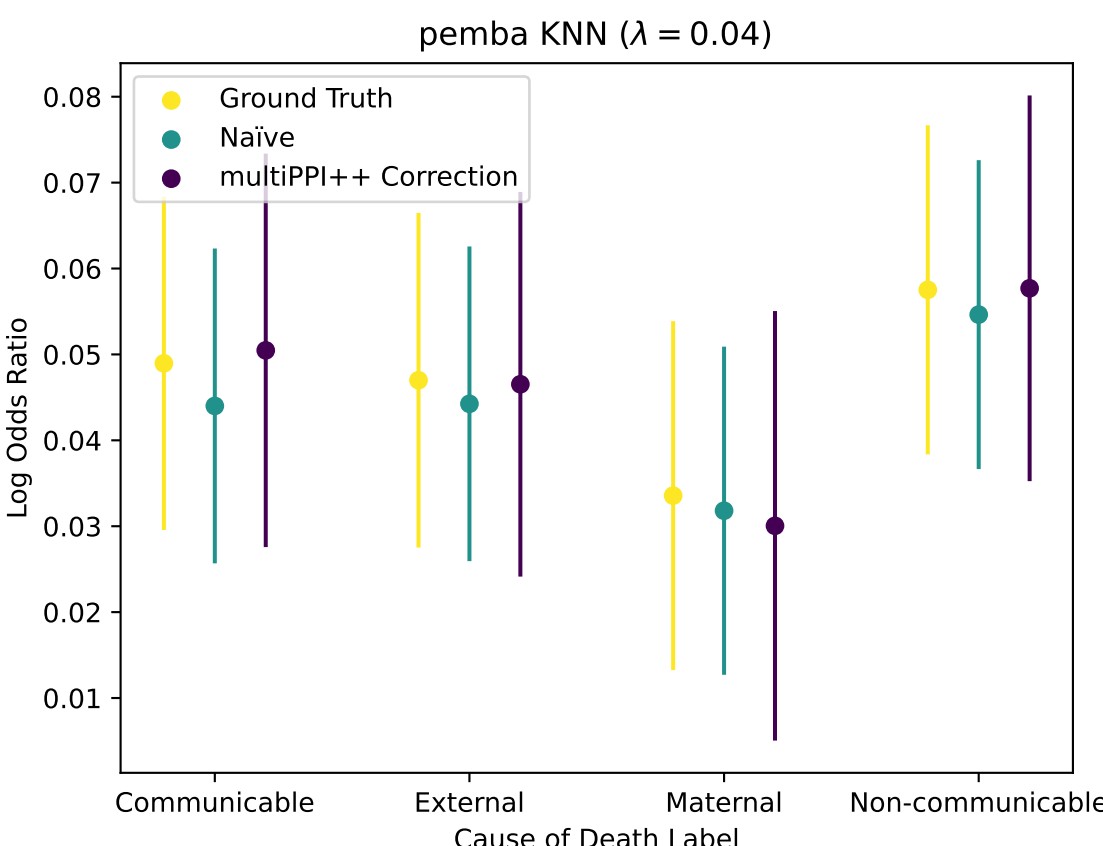

Figure 34: Stratified 80/20 Split: Site/Model 22

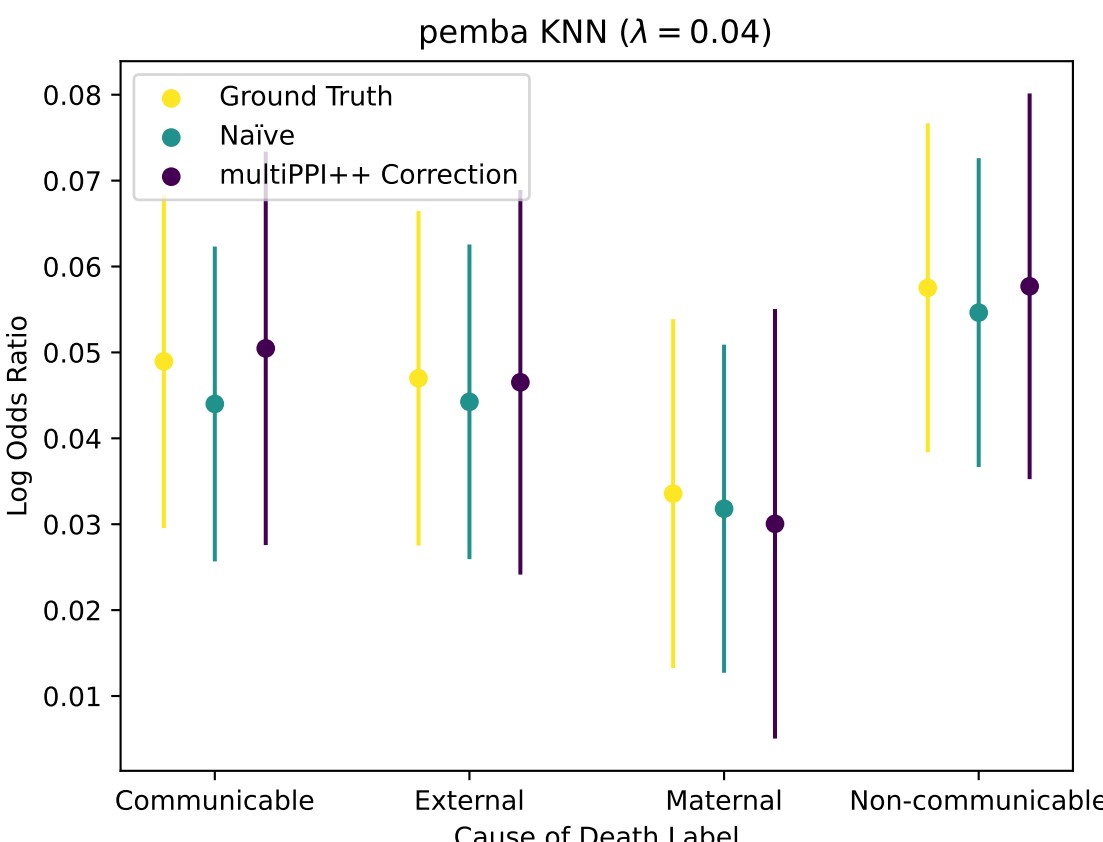

Figure 35: Stratified 80/20 Split: Site/Model 22

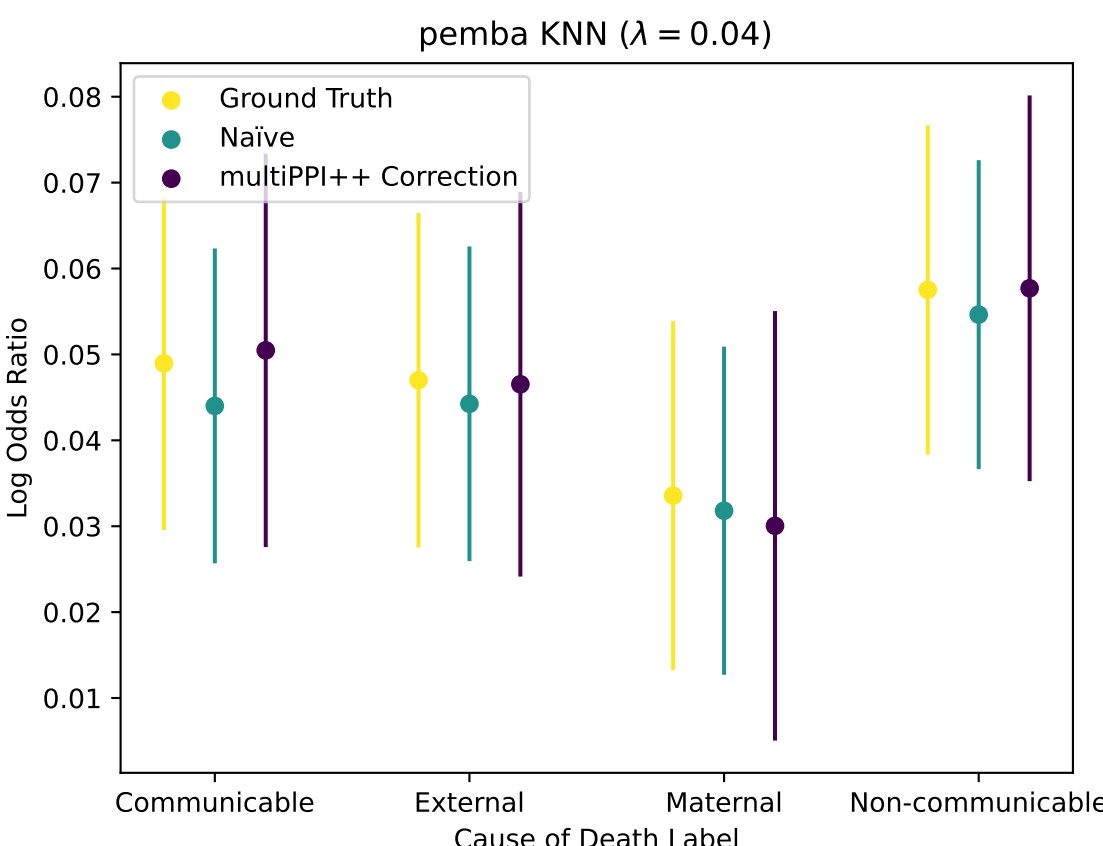

Figure 36: Stratified 80/20 Split: Site/Model 22

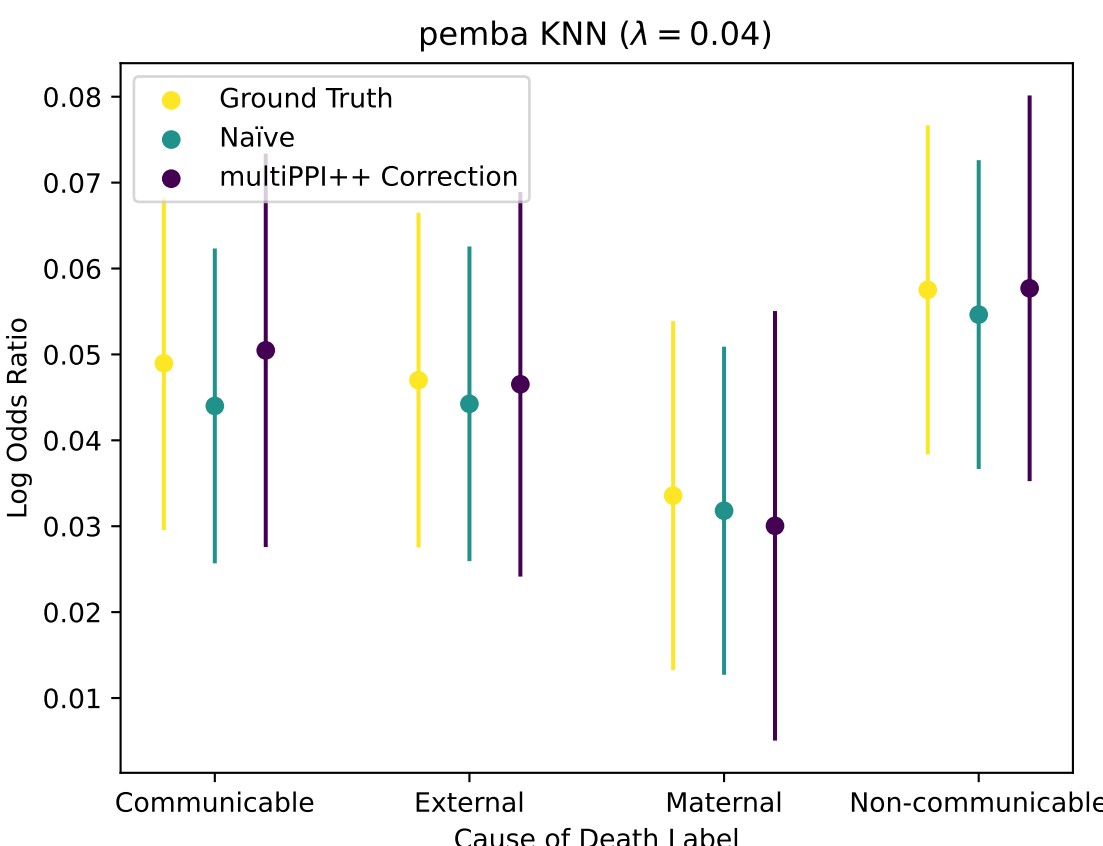

Figure 37: Stratified 80/20 Split: Site/Model 22

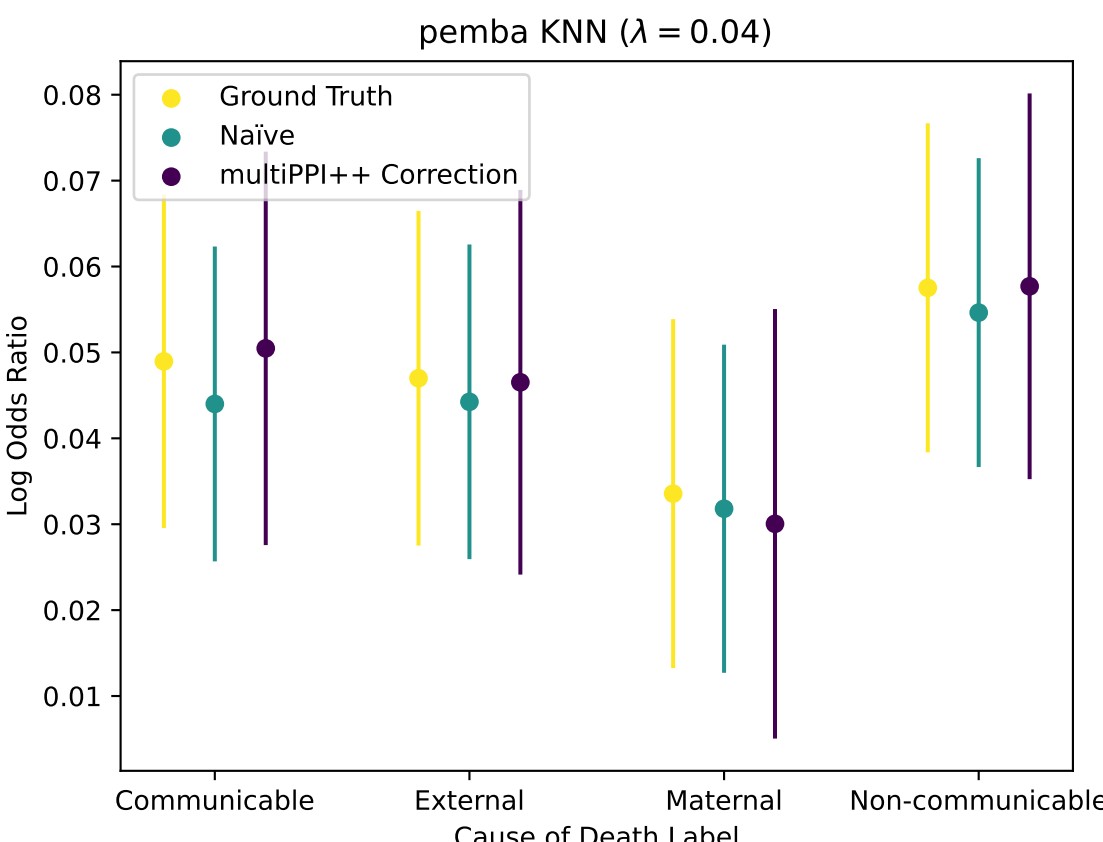

Figure 38: Stratified 80/20 Split: Site/Model 22

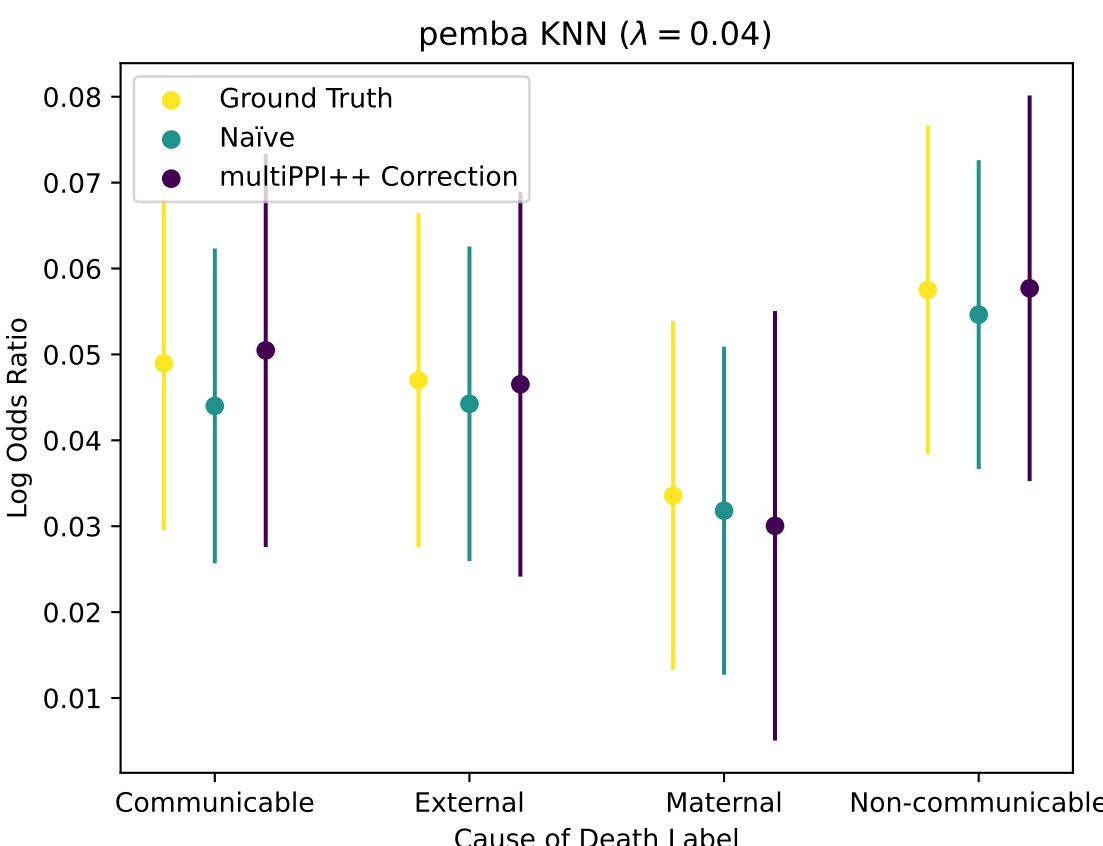

Figure 39: Stratified 80/20 Split: Site/Model 22

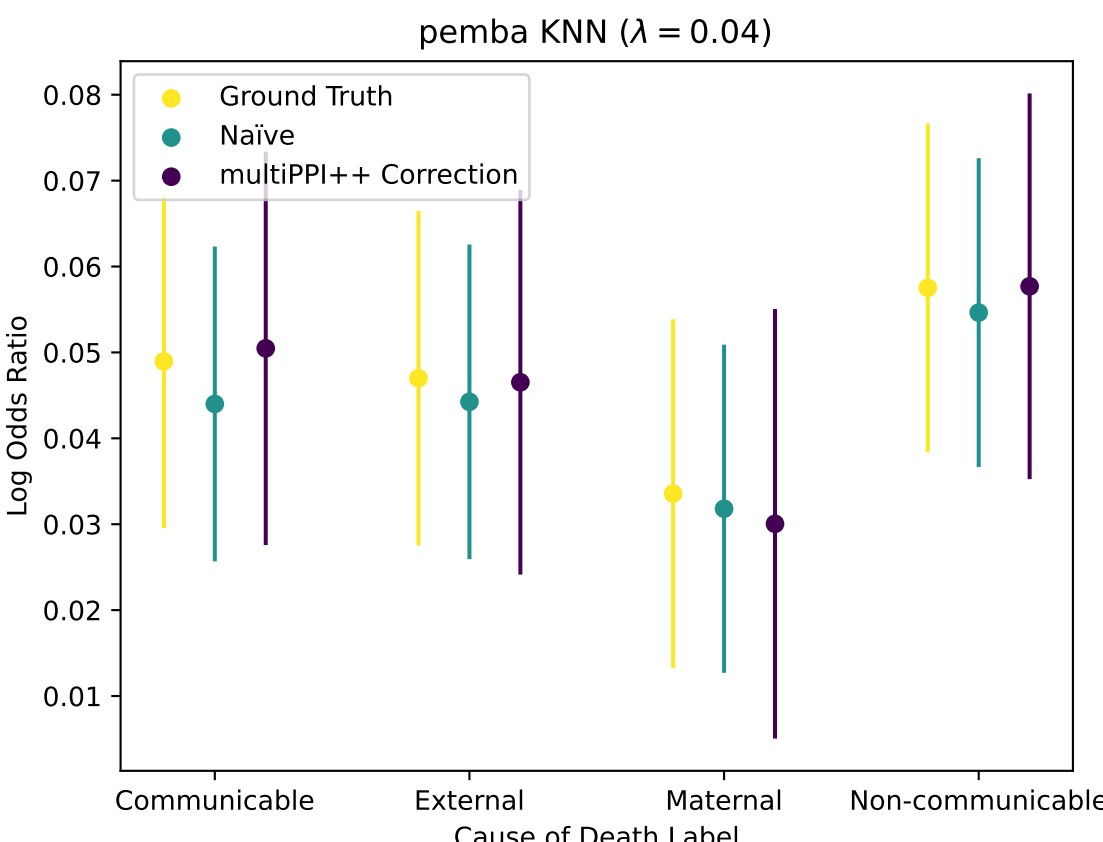

Figure 40: Stratified 80/20 Split: Site/Model 22

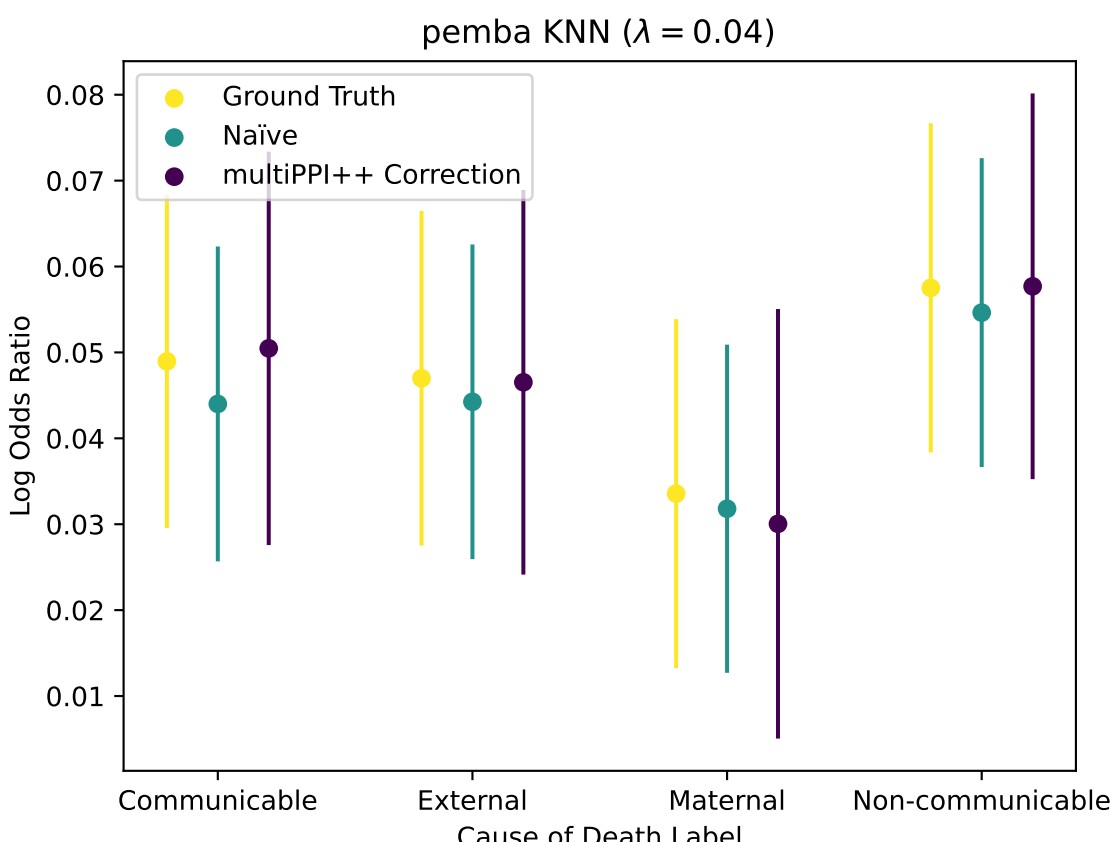

Figure 41: Stratified 80/20 Split: Site/Model 22

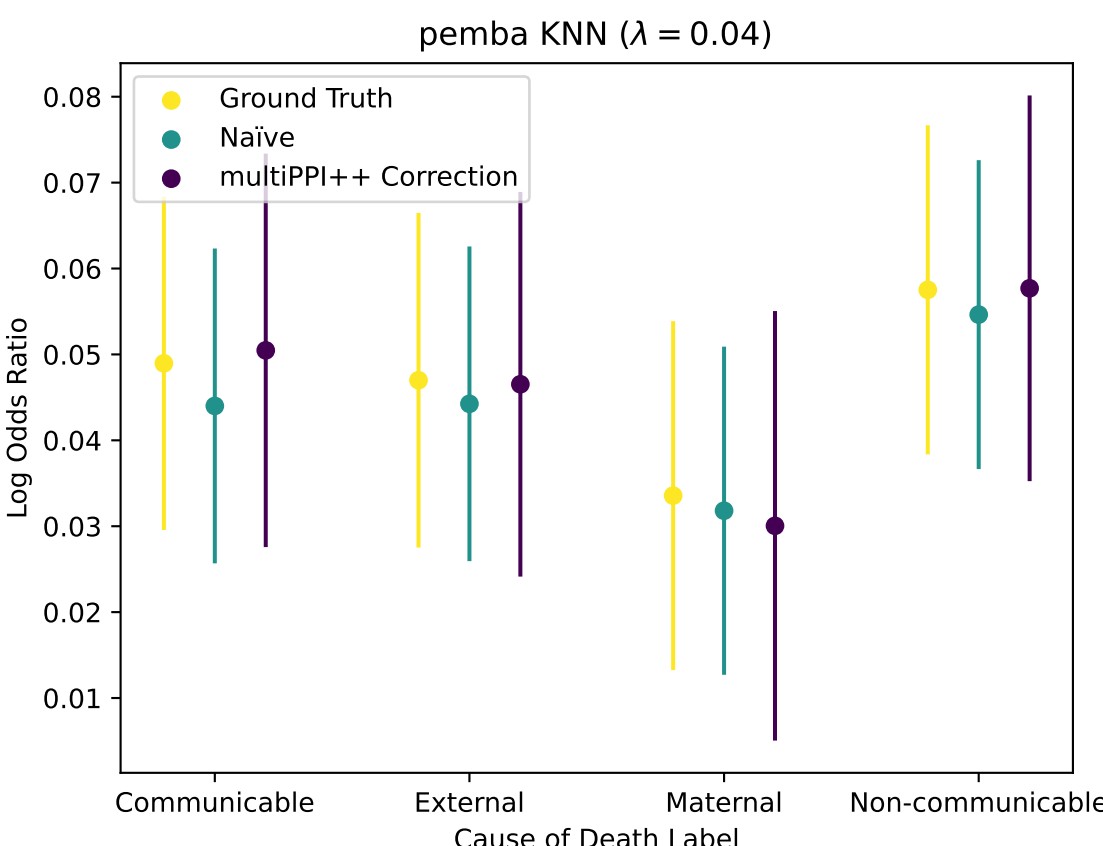

Figure 42: Stratified 80/20 Split: Site/Model 22

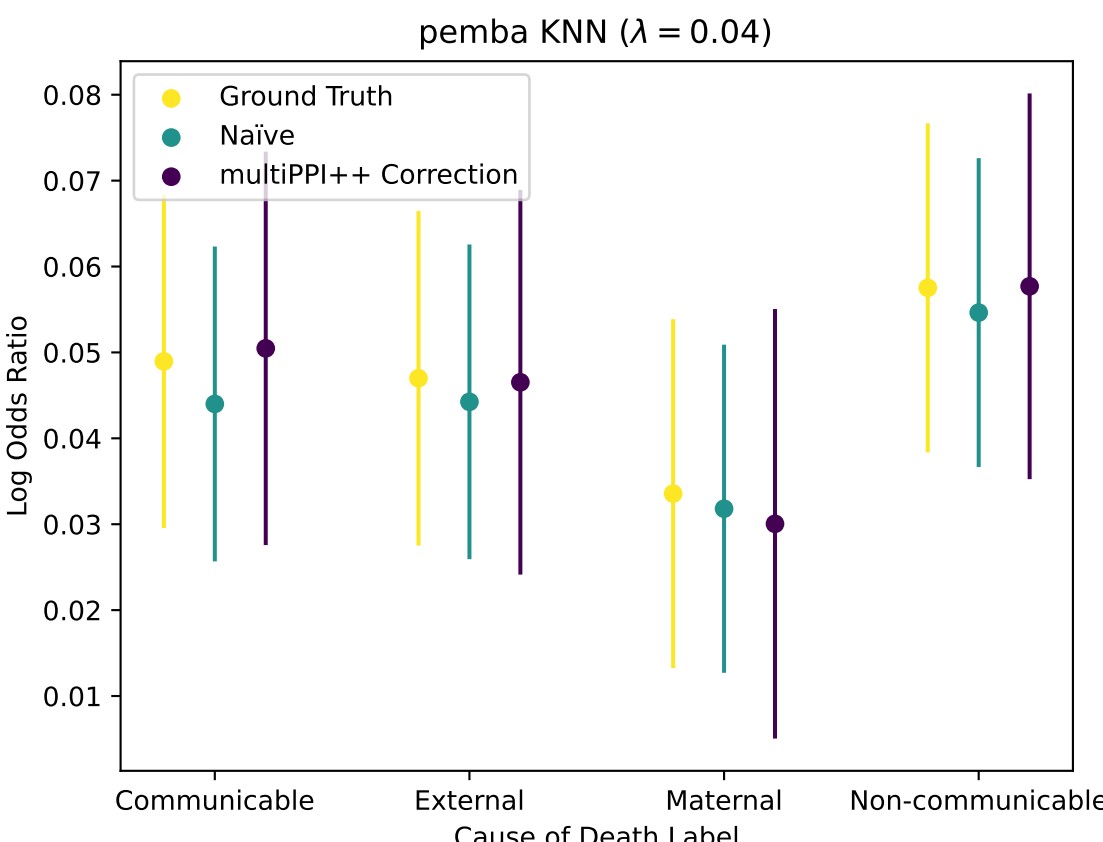

Figure 43: Stratified 80/20 Split: Site/Model 22

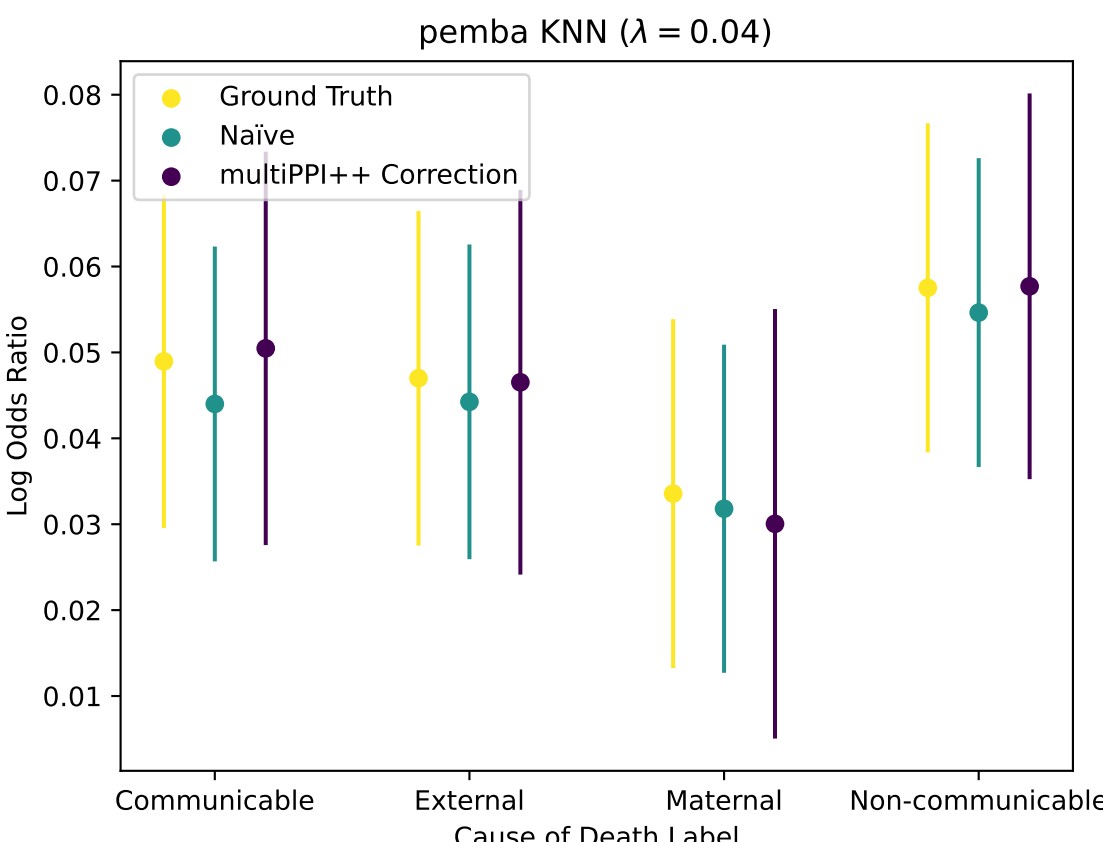

Figure 44: Stratified 80/20 Split: Site/Model 22

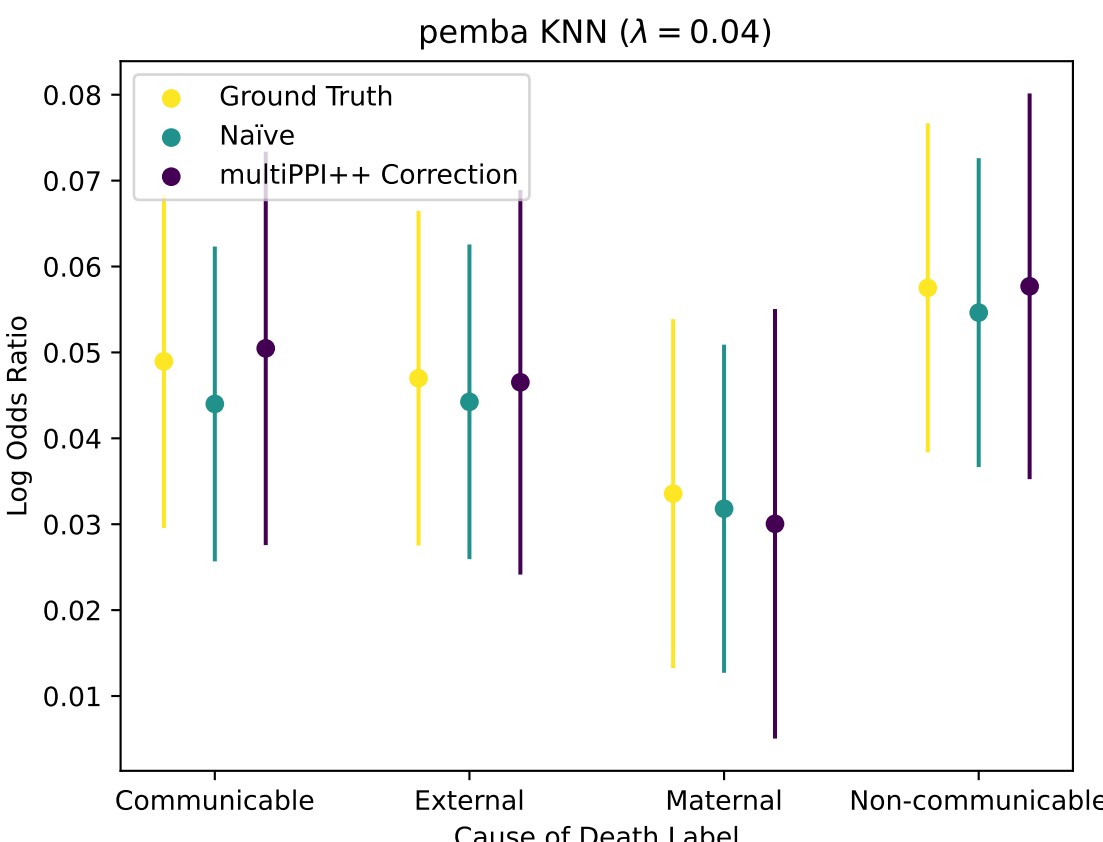

Figure 45: Stratified 80/20 Split: Site/Model 22

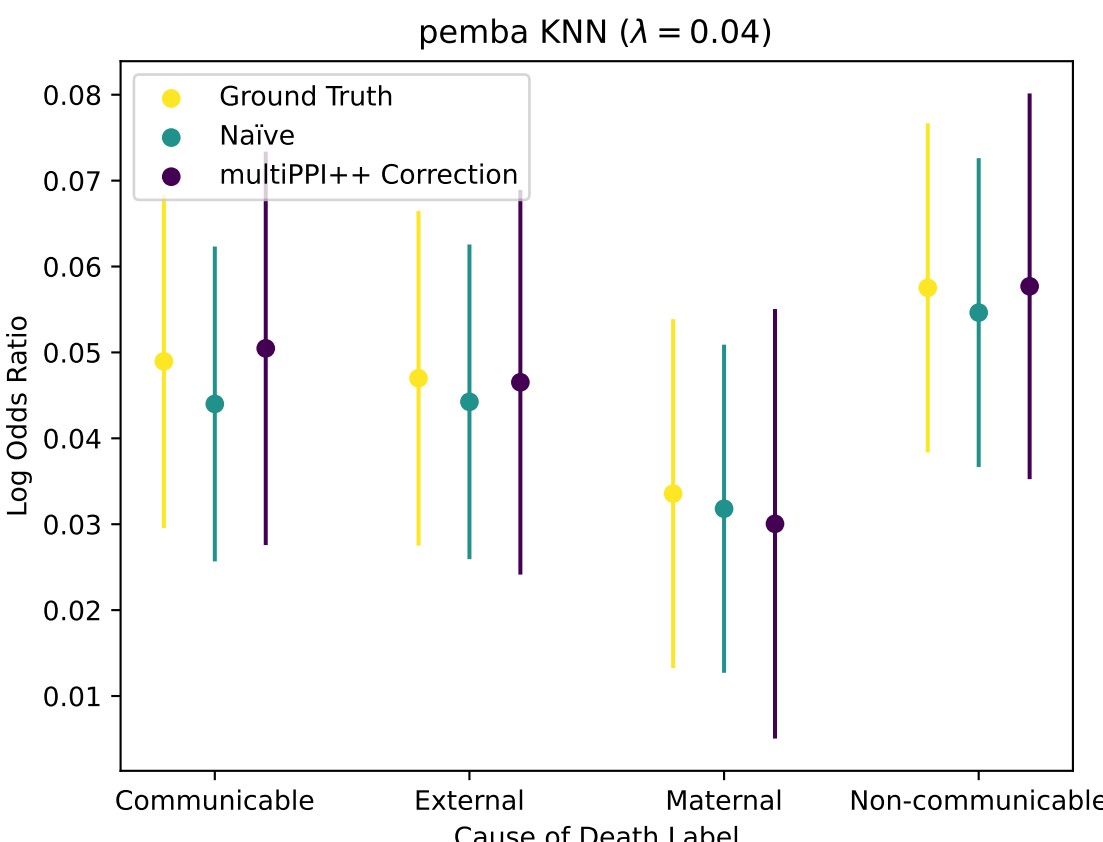

Figure 46: Stratified 80/20 Split: Site/Model 22

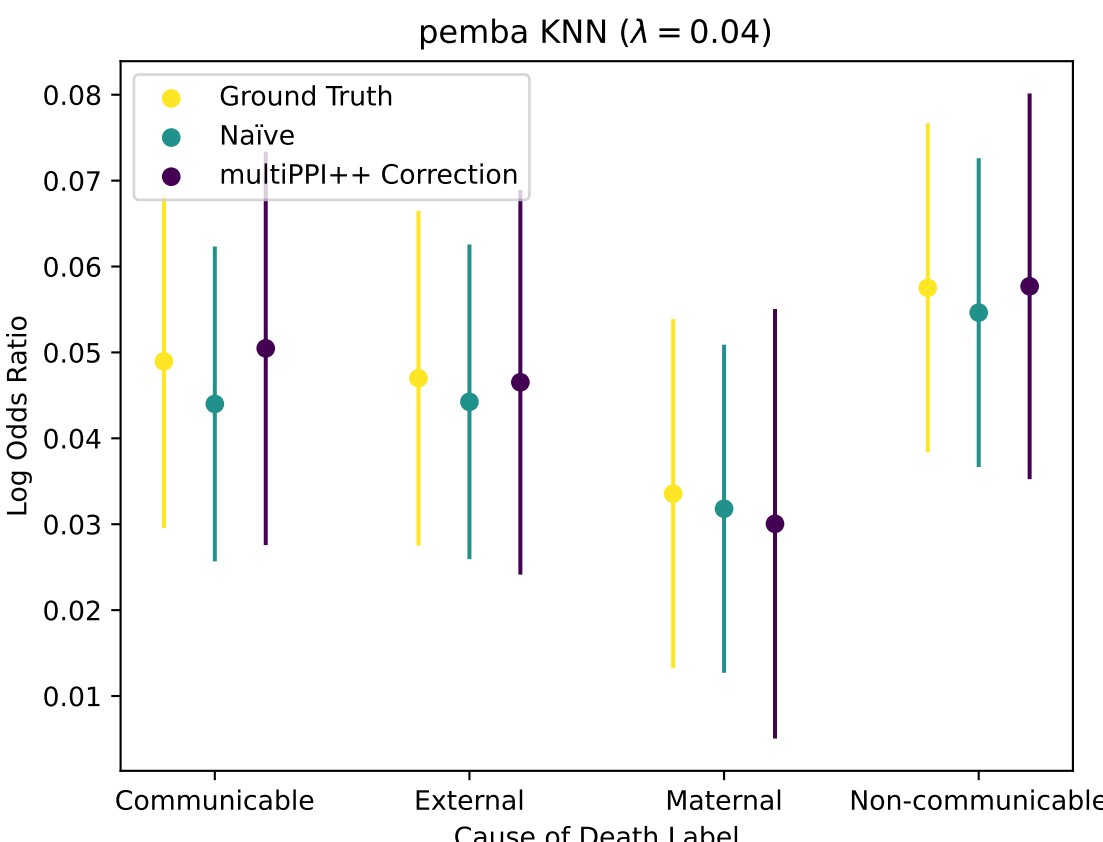

Figure 47: Stratified 80/20 Split: Site/Model 22

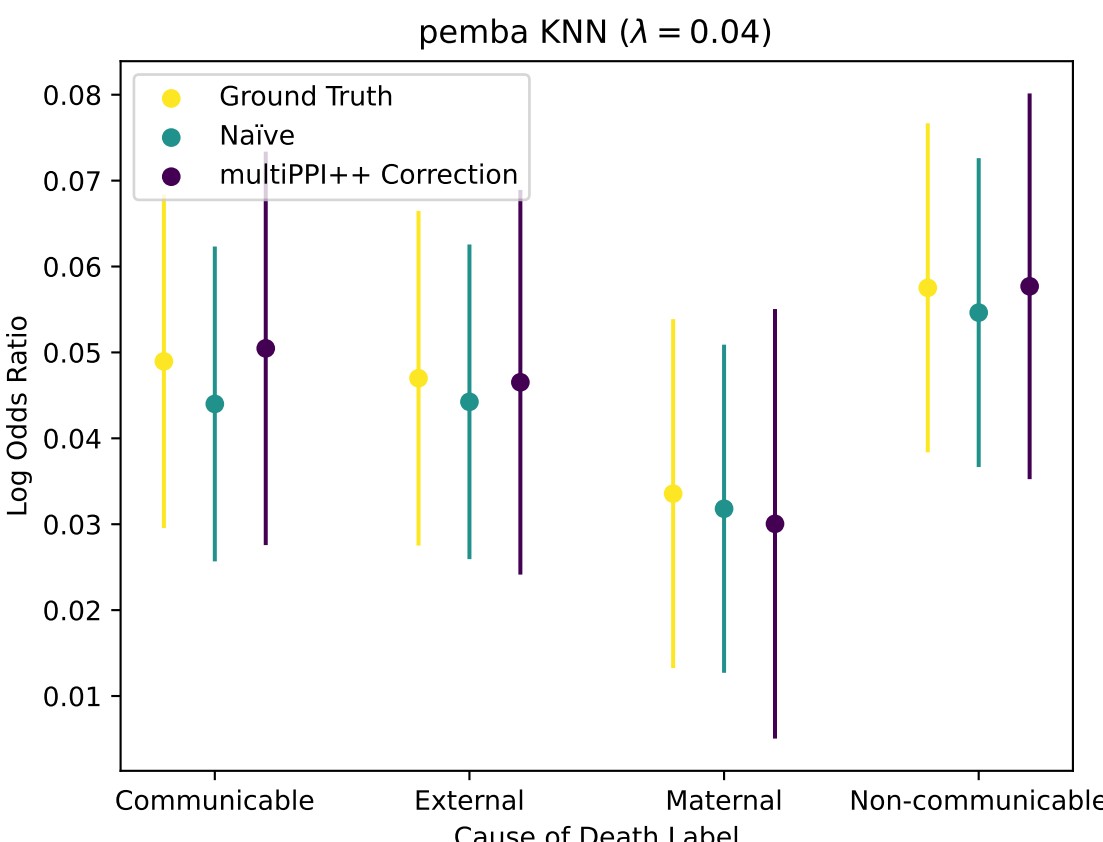

Figure 48: Stratified 80/20 Split: Site/Model 22

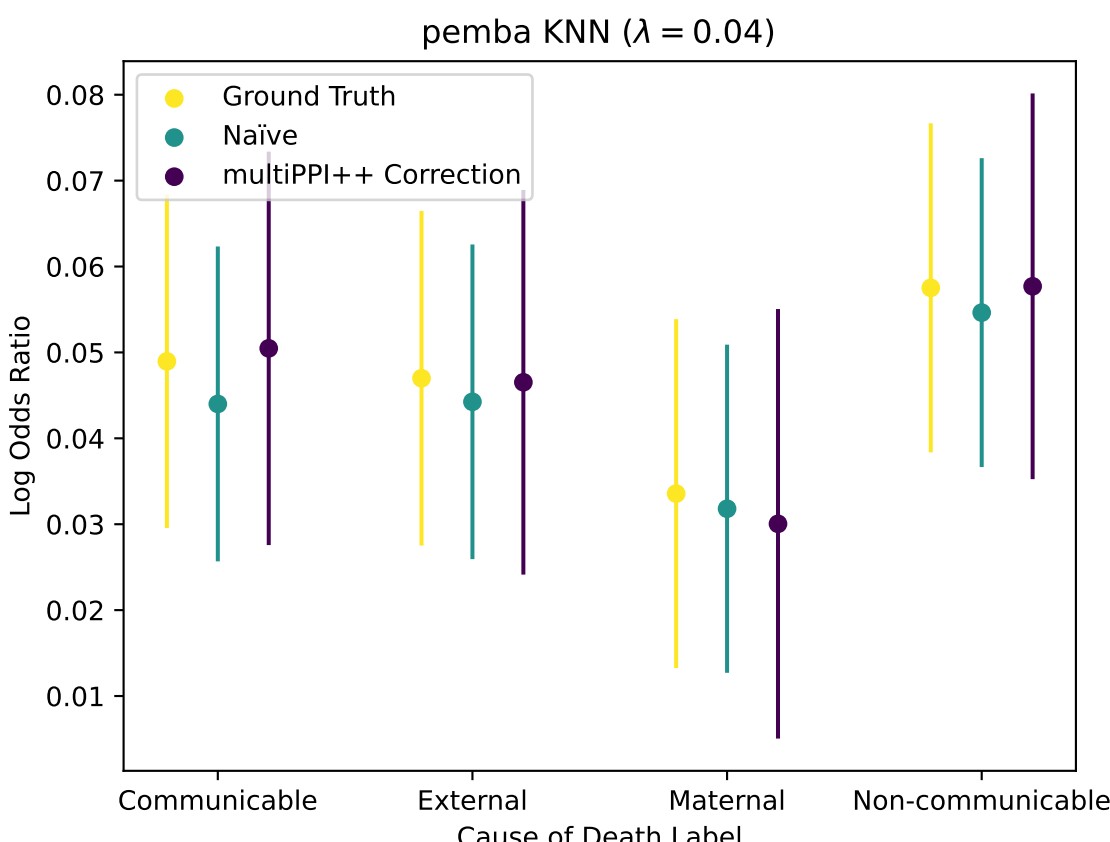

Figure 49: Stratified 80/20 Split: Site/Model 22

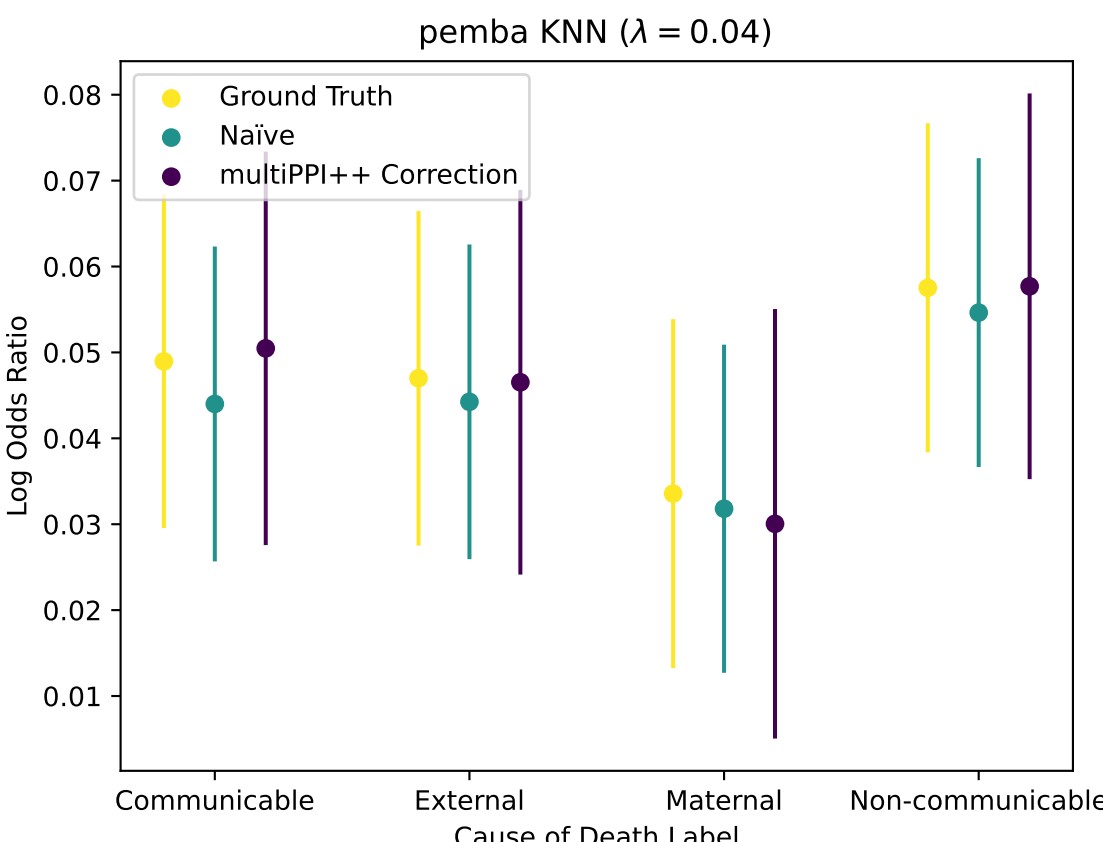

Figure 50: Stratified 80/20 Split: Site/Model 22

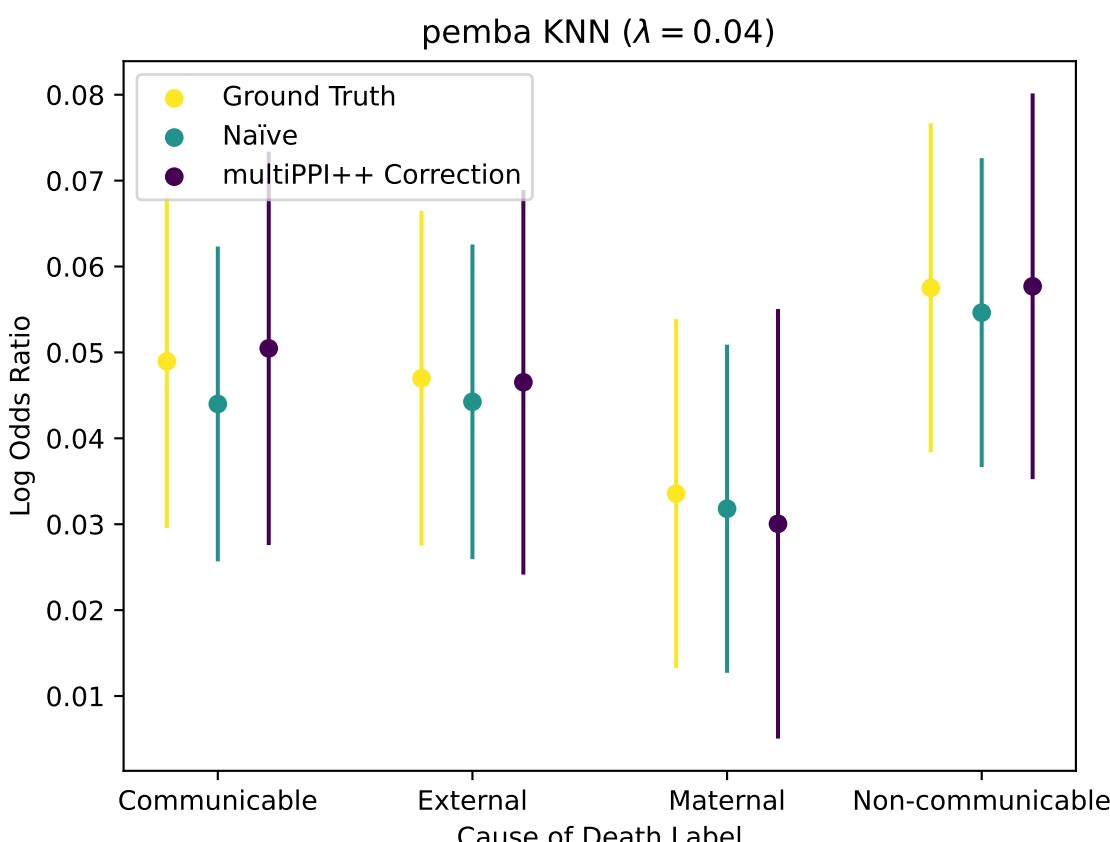

Figure 51: Stratified 80/20 Split: Site/Model 22

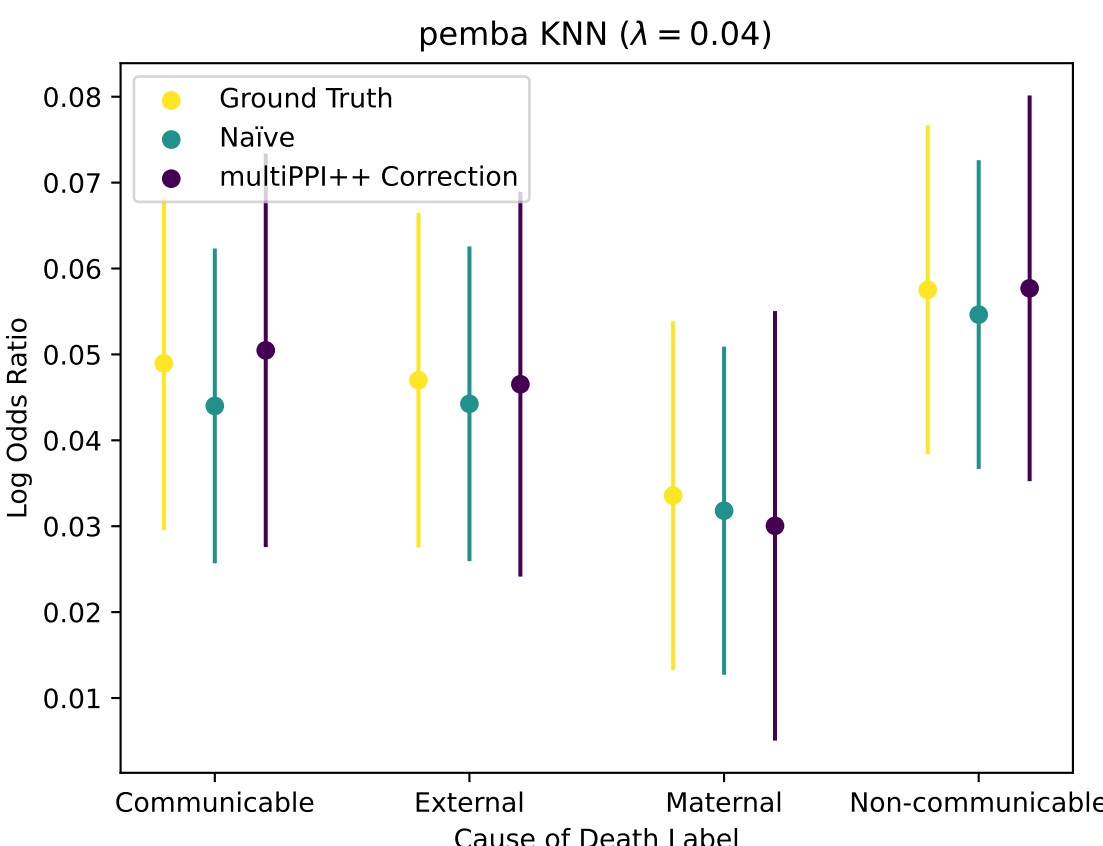

Figure 52: Stratified 80/20 Split: Site/Model 22

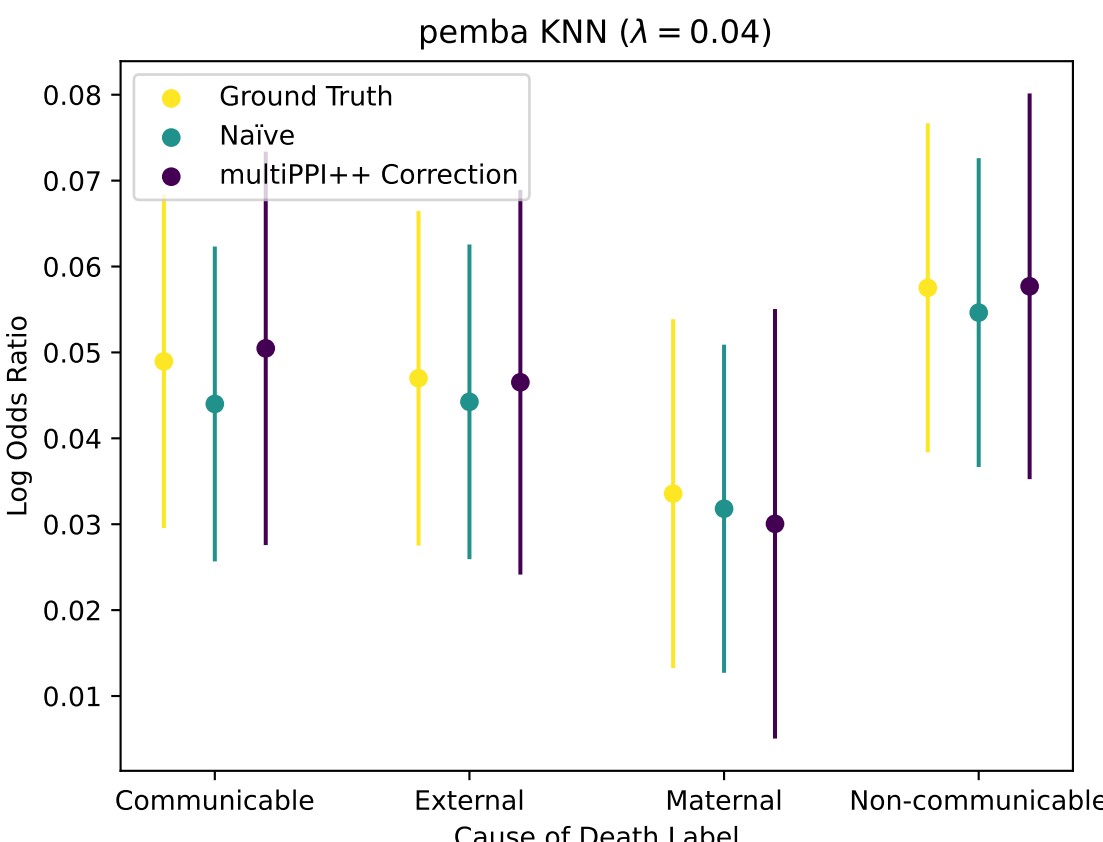

Figure 53: Stratified 80/20 Split: Site/Model 22

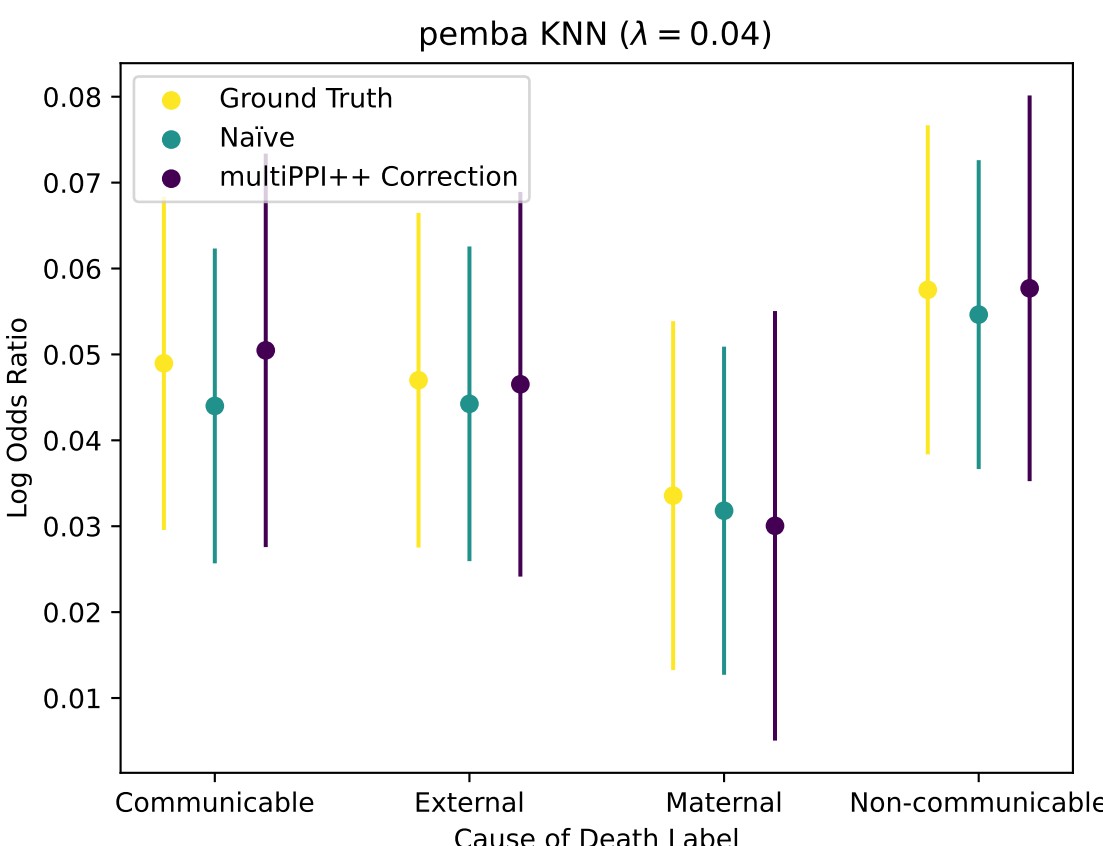

Figure 54: Stratified 80/20 Split: Site/Model 22

## B   Ethics Statement

As this paper covers the study of the lives and deaths of real human beings, the ethical aspect of this investigation was considered of paramount importance. First, to ensure that any potential personally identifiable information was not mishandled, multiple layers of data security were employed, the primary means being the use of a secure Microsoft Azure cluster for the running of local models. Data concerns are always paramount. We are fortunate to be working with de-identified data, however, when working with more sensitive data there are a number of open-source LLMs which can be run locally (Llama-3, meditron, etc) to ensure safe and ethical research practices. Additionally, many institutions, public and private, buy access to local OpenAI servers. Our use of these methods is illustrative, aiming to lay a foundational understanding of how to perform valid inference with NLP predictions. When communicating with outside organizations such as OpenAI's GPT models, we ensured that all data set were from the publicly available PHMRC data, which has been previously deidentified. Finally, in accordance with the importance of studying a sensitive topic such as VAs deserve, we have done our utmost to ensure that our study's objectives and tools utilized, to the best of our abilities, gives a fair, and respectful analysis of subject matter in question so that its results may be used for the furtherment of public health goals.

## C   Reproducibility Statement

To ensure the reproducibility of our results, we have published our code repository with our entire pipeline - NLP training, predicting, inference and visualizations - to Github. Please direct questions about data or modeling via email to `avisokay` or `fansx` at `uw.edu`. All data is available in the data folder in the Github repository.

