# OpenReview forum: "From Narratives to Numbers: Valid Inference Using Language Model Predictions from Verbal Autopsies"
_colmweb.org/COLM/2024/Conference — COLM_

### Official Review · Reviewer_VDYf · 2024-05-05

**Rating:** 4
**Confidence:** 4
**Ethics Flag:** 2

**Summary:**

This paper presents a methodology for deriving statistically sound inferences from causes of death predicted using verbal autopsy (VA) narratives and modern NLP techniques. Their approach, termed multiPPI++, is an extension of prediction-powered inference adapted for multinomial classification scenarios commonly encountered in public health research. Despite the variability in model accuracies, the multiPPI++ method consistently reconciles predicted causes of death with a small subset of verified ground truth data Results suggest that inference correction is important in health-based decision making

**Ethics Concerns Details:**

The issue of sending THIS dataset to OpenAI is addressed in the Ethics section, but this simply would not be legal on ACTUAL healthcare data across many jurisdictions

**Questions To Authors:**

-	Is there a concern about sending healthcare data to OpenAI’s servers, that could limit the use of this tool broadly? You had some flexibility given your dataset, but this wouldn’t this contradict data legislation for actual healthcare data?

**Reasons To Accept:**

-	The multiPPI++ method is somewhat novel approach to multinomial classification and may be appropriate for the complex data environments common in public health
-	The empirical validation is well supported by the very extensive supplemental material, although much of the detail in that (overly large) section does not make its way back into the paper itself
-	This is an important area of work, especially for low-resource settings where most deaths occur outside hospitals an few autopsies are performed.

**Reasons To Reject:**

-	There has been a lot of work over the past decade on modern NLP methods used to assess verbal autopsies, and relatively little of that was cited. Rather, the literature review somewhat superficially touches on each of the component topics separately (e.g., NLP in the whole, verbal autopsies in the whole) but not their previous confluence. Indeed, despite the mere fact that there is some novelty (as mentioned above), this is broadly a slight evolutionary step.
-	The empirical validation in the paper is somewhat light, and the rationale could be improved. In particular, the comparison of archaic methods like KNN and SVM to GPT-4 provides a massive gulf that one would expect to be filled by modern pre-trained language models that can be fine-tuned and run on-premises.
-	Little attention is paid to typical issues with data of this type, especially overfitting or other forms of poor generalization.
-	There is little justification for the complexity of the multiPPI++ process, nor are tradeoffs (e.g., between complexity and performance improvement) explored. Better ablations may have elucidated this.

---

> ### Author Rebuttal · Authors · 2024-05-31
>
> We thank the reviewer for raising these concerns and providing constructive feedback. We agree that in our review of prior work we should be more thorough in citing the confluence of VA and NLP. We intend to include more specific examples such as Manaka et al. 2022a and 2022b. These authors use BERT and ELMo for feature extraction from VA narratives to enhance COD classification tasks but do not comprehensively address the COD class imbalance challenge. We will note that this is a difficult problem in this domain that our method helps to address.
>
> Regarding the gap between BoW methods and GPT-4, it is our intent to show that our method multiPPI++ is model-agnostic, and applicable across a range of NLP models. This is an important point because practitioners could feasibly use any number of NLP methods, sophisticated or archaic, and our method remains robust and adaptable to models of varying performance.
>
> Echoing our response to Reviewer 2 regarding class imbalance and overfitting, we point out that we conducted a sensitivity analysis (see Figures 29-54 in A.6), where we used one approach to correct for class imbalance by training the model on stratified data, with an 80/20 split for each class to achieve balance w.r.t COD. The results were not substantively different from our main results, so we showcase an example where the NLP model is not perfect, but multiPPI++ still obtains valid statistical estimates. To clarify our attention to this important concern, we will incorporate more discussion, as well as comparisons to state of the art methods (e.g., SMOTE).
>
> To expand on the multiPPI++ complexity and tradeoffs, we will add clarification to emphasize that MultiPPI++ does not change NLP prediction performance, it only affects the accuracy of the downstream statistical estimates and appropriately widens associated confidence intervals to account for the additional originating from the NLP model. We will take care to emphasize this point more clearly.
>
> Data concerns are always paramount. We are fortunate to be working with de-identified data, however, when working with more sensitive data there are a number of open-source LLMs which can be run locally (Llama-3, meditron, etc) to ensure safe and ethical research practices. Additionally, many institutions, public and private, buy access to local OpenAI servers. Our use of these methods is illustrative, aiming to lay a foundational understanding of how to perform valid inference with NLP predictions.

---

> > ### Comment · Reviewer_VDYf · 2024-06-05
> >
> > Thank you for the response. I agree with various elements of your rebuttal, and you touched on relevant aspects of my review. I should note that although a model-agnostic model _can_ run on older methods (and it is still good that you worked with them), this does not mean that you should _only_ (or mainly) focus on those older methods.

---

> > > ### Author Response · Authors · 2024-06-07
> > > **Follow up to rebuttal response.**
> > >
> > > Thank you, this is a very good point. To add more information that is relevant for VA on modern LM techniques, we propose to add more discussion about the prompting.
> > >
> > > We will add the following discussion of our prompt engineering in our section 3.1.
> > >
> > > “We faced a series of challenges in engineering this prompt, namely, coercing the LM to consistently return outputs constrained to our exact list of output classes [communicable, non-communicable, external, maternal, aids-tb]. Earlier iterations of the prompt yielded responses such as “The patient died due to external causes.” While perhaps semantically correct, validating outputs such as this at scale is intractable and would require additional steps (regex, fuzzy string matching, etc.). Instead, we found that including our desired output classes between html tags <options> </options> and explicitly asking the LM to return only a choice from the list of options in the prompt improved results considerably. Future work may include few shot prompting with select domain-specific examples or retrieval augmented generation from a larger database of narratives.”
> > >
> > > To conclude this section we will add the following sentence.
> > >
> > > "We include outdated NLP approaches like the BoW representation classification models in our analysis because they are cheap and quick options, however we will focus our attention on interpreting the results from the GPT-4 modeling as it represents the current state of the art in language modeling and produces the most accurate results in this domain."

---

### Official Review · Reviewer_wuA5 · 2024-05-08

**Rating:** 5
**Confidence:** 3
**Ethics Flag:** 1

**Summary:**

The authors present a benchmark for predicting the Cause of Death from Verbal Autopsies and introduce a method, called multiPPI++, to identify quantitative factors responsible for the cause of death, such as age.

The experimental setup is constrained by geographically-dependent distributions and investigates valid inference with a mix of ground truth and predicted labels.

While experiments are sound and multiPPI++ are interesting, the conclusion and contributions do not appear significant enough. The contributions are focused on an NLP-agnostic statistical method rather than advancing LLMs (as a field).

**Reasons To Accept:**

The problem tackled is interesting and addressed with a consistent experimental setup in which results are correctly analyzed.

The exploratory data analysis helps anchor the challenges of geographically-dependent Cause of Death distributions.

The use of Language Models, in comparison with older NLP techniques, to study inference and prediction of the Cause of Death from Verbal Autopsy Narratives, represents an opportunity to investigate an important topic.

**Reasons To Reject:**

Originality is limited. The crux of the novelty in the proposed method is to adapt a published method called PPI++ to settings of multinomial classification.
The paper is not well suited for a conference on Language Models but rather adapted for a statistics venue.

Some claims are either too vague, not corroborated or not explained:
- "Angelopoulos et al. (2023b) [...] 's model has overparametrization issues."
- "GPT-4’s predictions matched the performance of the best examples from the VA literature."

The presentation could be adapted to improve the clarity of 1) the motivation and 2) the method.
Also, the paper could benefit from a dedicated section focusing on Related Works.

---

> ### Author Rebuttal · Authors · 2024-05-31
>
> We thank the reviewer for this feedback and agree that we can do more to emphasize why this work is an important contribution to advancing the field of LLMs. While it is true that we are not proposing new NLP algorithms, NLP methods are used in an extremely wide range of settings, often with the goal of performing downstream inference. Our multiPPI++ method enables valid inference when outcomes are predicted using an NLP model and, thus, enhances the utility of NLP predictions. We believe this is a contribution to the field as it is an important step in the process of leveraging NLP to make sound, data-informed decisions.
>
> We thank the reviewer for bringing this to our attention and agree that these claims are not sufficiently supported in our initial submission. We will elaborate on how our parameterization overcomes the issues which arise in Angelopoulos 2023b. We will also include citations to the leading prediction accuracies in the existing literature and provide a direct comparison to our NLP prediction accuracies using text narratives. These clarifications will specify and substantiate our claims you identified above.
>
> We thank the reviewer for their valuable comments highlighting all of the above ways we can improve the clarity, scope, and motivation of this work. In addition to what we have outlined above, we will include a Related Works section to our discussion to emphasize what motivates our work and showing how our contribution is situated in the language modeling literature.

---

> > ### Comment · Reviewer_wuA5 · 2024-06-06
> >
> > Thanks for your rebuttal.

---

### Official Review · Reviewer_hhrX · 2024-05-11

**Rating:** 6
**Confidence:** 3
**Ethics Flag:** 1

**Summary:**

This paper applies an array of multi-class classification approaches to cause of death prediction from verbal interview transcriptions with the bereaved family (a KNN, NB, SVM BOW approach), a fine-tuned BERT model and GPT-4.
The authors further claim to propose the adaptation of PPI++ to multiclass logistic regression, in order to perform domain adaptation for a model trained on domain A (with labelled samples available) to domain B (with only unlabelled elements).
The authors independently evaluate the approaches in a classical fully-labelled setting. Without any class-imbalance mitigation strategy, most of the BOW models and BERT overfit on the majority class prediction. Only GPT-4 doesn't suffer from this problem and mainly only confuses Communicable with non-communicable causes. No error analysis is provided to truly understand why or whether prompt adaptation strategies could solve the problem.
As for multiPPI++, which is used to enable domain transfer, only the distributions of corrected  confidence estimates on the loss ar4e provided, this absolutely doesn't evaluate whether it helps for domain transfer.

Normally multinomial classification means the same thing as multiclass classification, which Angelopoulos et al. (2023 b) https://arxiv.org/pdf/2311.01453, already define on page 7 section 3.1.
Do the authors mean multilabel multi-class classification?  But then is the only adaptation computing the mean of the first terms across labels ? PPI++ involves many prerequisites w.r.t. the loss function, does your modification preserve these prerequisites? Particularly the stochastic smoothness? Maybe it's obvious for someone more mathematically adepts, but it feels like this is the sort of thing that's extremely important to assess the validity of what's proposed.
I feel like your modification assumes the multiple outcomes are independent from each other, is this a reasonable assumption? I am not convinced it's that simple...

**Questions To Authors:**

A.1 ICD-10 COD Classification > Better to include the actual ICD-10 codes

"We then applied three
supervised classifiers to predict COD using the BoW representations and ground truth cause of death
labels: Support Vector Machine (SVM), K Nearest Neighbors (KNN), and Naive Bayes (NB). SVM
was implemented with a third-degree linear kernel and C = 1.0, while KNN used cosine distance and
9 nearest neighbors, chosen by incrementing from K = 3 until accuracy saturation. The BERTBASE
encoder was configured with five transformer layers, to include two input layers of size 256, a main
layer of size 768, an intermediate layer of size 512, and a fine-tuning output layer of size 5. As the VA
narratives in our data were de-identified, we were able to further assess the predictive performance
of OpenAI’s GPT models (GPT-3.5, GPT-3.5-turbo, GPT-4, and GPT-4-32K Biswas, 2023)" > Was there any parameter estimation performed? did you have an a priori rationale to select these specific parameters?



F1-Score > Can you detail how you are computing the F1 score? In multinomial classification you would compute individual F1 scores for each class, the mean of class-wise F1-scores wouldn't be particularly meaningful...
Even then, we would perhaps like additional information (e.g. what the distribution of the of the scores is. In that sense the confusion matrices are much more informative.


Besides for GPT-4, it looks like all the models are overfitting and predicting the majority class! Something must be done to tackle class-imbalance, that's likely the reason why most of the approaches perform poorly (including BERT!)
Why isn't the confusion matrix from BERT also represented? Are you trying to hide issues that you may have had?


"Figure 6 displays point estimates and 95% confidence intervals for the ground truth, Naive
and multiPPI++ corrected inference" While it's a nice visualization, what meaningful information does this bring for the analysis?
Not really commented, what's the meaning of those difference? Does it have any significance in terms of domain expertise?

Does adding multiPPI++ improve or degrade performance? Why don't you report of the task performance for all three scenarios? You can train with the predicted label augmentation and evaluate only on full labels.
If this impreoves performance it's certainly a good indicator of the effectiveness, idependently of whether the uncertainty estimates of multiPPI++ better correspond to the true uncertainty estimates.


Most of the text in A2 is a verbatim copy of the text in (Angelopoulos et al., 2023a;b), if you are quoting, you should clearly say so, or just telle readers to read the appropriate section in the original papers.
Even if it's in the appendinx, this is wrongful attribution.


If multiPPI is a contribution of the paper, then it should be described adequately in the main body of the paper. Given the evaluation provided, I don't see any demonstration of the concrete usefullness of multippi++ (no because there isn't any, but because this is not evaluated).


I am missing an error analysis, particularly for models that do not overfit on the majority class, as we could for example understand why communicable and non-communicable are so often confused by GPT-4.


Why is algoritm 1 in the annex if it's cited in the body of the text, that's a way of cheating the page limit in my opinion.

As far as I see in (Angelopoulos et al., 2023a;b) there is a specific "stochastic smoothness" condition on the loss function, for this method to work, besides the iid requirement. Did you actually verify any of these requirements to determine if the method is applicable at all?

**Reasons To Accept:**

The task explored is interesting from the perspective of biomedical informatics, as it's something that's not typically not addressed much.
In theory the proposed extension would be a very interesting contribution (if done right, with the right assumptions).

**Reasons To Reject:**

The experimental validation of the various approaches without multiPPI++ is very incomplete, we need a much more detailed report, with a complete error analysis and discussions on prompting strategies. Class-imbalance must also be addressed, otherwise the behaviour of the simple baselines is really to be expected.

The proposed extension of PPI++ seems to be made under questionable simplifying assumptions that do not seem realistic (independence of selected subset of outcomes). The requirements of PPI++ regarding the modified loss aren't discussed and there is nothing that indicated whether they still hold (even if it's supposed to be obvious, some readers might need to explicitly see it stated with at least an informal justification.

I feel like the paper needs to focus on one contribution: the application task (with a proper evaluation, including domain transfer ability) or the more fundamental extension of multiPPI++ (with proper proofs and mathematical justifications!).

---

> ### Author Rebuttal · Authors · 2024-05-31
>
> We thank Reviewer 2 for their detailed review and helpful comments. Regarding domain transfer, a primary aim of this work is to illustrate how predicted outcomes from an NLP model can be adjusted to yield accurate statistical parameter estimates even when a domain transfer has occurred. Additionally, an alternative view of this problem would be to account for ‘covariate shift,’ i.e., when the underlying distribution of our predictors differs across datasets. There has been discussion of this in Angelopoulos et al. (2023), which can readily be adapted to this work.
>
> In section A2, we will direct readers to the specific section of Angelopolulos et al rather than quoting their description.
>
> We will add ICD-10 Codes to the table in A.1.
>
> We compute f1-score using the standard sklearn implementation and show confusion matrices in Figure 5.
>
> While we include our prompt in Figure 3, we agree that we do not sufficiently discuss our strategy. We will elaborate on the prompt engineering and associated error analysis in that section.
>
> Regarding class imbalance, in a sensitivity analysis (see Figures 29-54 in A.6), we used one approach to correct for class imbalance by training the model on stratified data, with an 80/20 split for each class to achieve balance w.r.t COD. The results were not substantively different from our main results, so we showcase an example where the NLP model is not perfect, but multiPPI++ still obtains valid statistical estimates. To clarify our attention to this important concern, we will incorporate more discussion, as well as comparisons to state of the art methods (e.g., SMOTE).
>
> As for the performance, Figure 6 demonstrates that multiPPI++ improves the accuracy of statistical estimates using a post-facto correction term. MultiPPI++ does not change prediction accuracy, it only affects the accuracy of the statistical estimates and appropriately widens the confidence intervals to account for the additional uncertainty. We will take care to emphasize this point more clearly.
>
> Regarding multinomial classification, Agelopoulos et al. 2023 provide a formula. We take that formula, reparameterize it, and develop an optimization algorithm. We will clarify that this does not affect the statistical properties of the estimates (the stochastic smoothness condition is satisfied, as $\nabla \theta$ is locally Lipschitz near $\theta^*$ for the problem.)

---

> > ### Comment · Reviewer_hhrX · 2024-06-06
> >
> > Thank you for your response, I have increased my rating to 6 following the response.
> >
> > If you give a precise account of changes proposed quoting text that will be added and where it will be added, this may allow me to raise the score further

---

> > > ### Author Response · Authors · 2024-06-07
> > > **Responding to follow up comment**
> > >
> > > Thank you for the suggestion, we will include these specific changes.
> > >
> > > In regards to the prompt engineering, we will include the following at the end of our section 3.1.
> > >
> > > “We faced a series of challenges in engineering this prompt, namely, coercing the LM to consistently return outputs constrained to our exact list of output classes [communicable, non-communicable, external, maternal, aids-tb]. Earlier iterations of the prompt yielded responses such as “The patient died due to external causes.” While perhaps semantically correct, validating outputs such as this at scale is intractable and would require additional steps (regex, fuzzy string matching, etc.). Instead, we found that including our desired output classes between html tags <options> </options> and explicitly asking the LM to return only a choice from the list of options in the prompt improved results considerably. Future work may include few shot prompting with select domain-specific examples or retrieval augmented generation from a larger database of narratives.”
> > >
> > > Regarding class imbalance, we will add the following after the first sentence in our section 4.2 to expound on this point.
> > >
> > > “As evidenced by the COD distribution displayed in Figure 2, we are dealing with considerable class imbalance within and between sites. We address this in a sensitivity analysis (see Figures 29-54 in A.6) where we train each model on a stratified sample with an 80/20 split for each class to ensure an equal proportion of training examples for each class. The results from this analysis are not substantively different from our main results where we do not account for the class imbalance in our modeling. Alternative resampling methods such as SMOTE, ADASYN or Tomek Links could be applied to correct for the imbalanced distribution of COD. However, even in the absence of such considerations, we show that multiPPI++ still obtains valid estimates in downstream inference using predictions from imperfect NLP modeling. This extensibility of our approach is one of the advantages of the multiPPI++.”
> > >
> > > Regarding our multiPPI++ performance, we will add this to our section 4.3 to give additional context in interpreting Figure 6.
> > >
> > > "Figure 6 demonstrates that multiPPI++ improves the accuracy of statistical estimates using a post-facto correction term estimated using relatively small amounts of labeled data. As this is applied after model predictions are made, multiPPI++ does not improve or degrade model accuracy. This procedure provides a correction for the biased statistical inference parameters and the deflated uncertainty intervals around them, given NLP produced estimates of likely COD from VA narrative predictions. This is a nuanced yet important distinction as these downstream statistical inference parameters are what describe the relationship between covariates, such as age, and cause of death distributions. This same intuition extends to other domains where the end goal is to use inference to guide decision making where NLP predictions are used in place of ground truth labels."

---

### Official Review · Reviewer_EkdW · 2024-05-15

**Rating:** 8
**Confidence:** 4
**Ethics Flag:** 1

**Summary:**

This paper develops an approach to perform *inference* on categorical predictions from a machine learning model applied to unstructured text (in contrast with prior work performing *prediction*) for the task of cause-of-death classification from narrative text.

**Questions To Authors:**

Can multiPPI++ be generalized to any model that uses a logistic / cross-entropy loss (eg. a frozen LLM with a logistic head)?

Could threshold-free metrics be reported in Section 4.2. Namely: the ROC curve and the ROC-AUC, and the precision-recall (PR) curve and the PR-AUC?

**Reasons To Accept:**

The paper considers an important task with the potential to significantly reduce human effort and improve decision-making accuracy. The paper is well-written, clear, does not over-claim, and is both theoretically and empirically principled.

**Reasons To Reject:**

The considered task, while important, is quite specific. A drawback of this paper is the focus on a single task, without empirical benchmarking of the proposed inference approach on multiple tasks.

The ethics statement is not specific enough for the nature of research being performed. The statement should warn readers of the variety of risks of using the proposed approach in practice, especially since the data includes human-provided narratives which might be biased or incorrect along with the biases embedded in machine learning models.

---

> ### Author Rebuttal · Authors · 2024-05-31
>
> We thank the reviewer for this feedback and recognize the importance of demonstrating the versatility and robustness of our contribution. We will highlight that our primary goal was to develop and validate an initial framework that could be further generalized and adapted to various tasks within the domain of cause-of-death classification. In particular, in our discussion about future work, we will add more details about how we plan to validate the approach with external data, as well as extend to other application areas such as  in analyzing clinical text.
>
> We agree that the ethical considerations related to our research need to be more clearly articulated. We will revise the ethics statement to (1) highlight the potential for biases in human-provided narratives (e.g, inaccuracies, incomplete information, or cultural, social, or personal perspectives of the individuals providing the narratives, (2) model-induced biases (e.g, bias from the methods which may embed biases present in the training data, and (3) misclassification (e.g., in particular from class imbalance, which we address in the appendix using an output-stratified bootstrap sample for our leave-one-out experiment. (See figure 10, for example)). Further, we will clearly state that in practice, ensuring the privacy and confidentiality of the individuals whose narratives are used is critical. We will reiterate that all of the data used in this study were anonymized, but other sources of narratives may preclude the use of pre-trained LLMs such as ChatGPT due to ethical concerns surrounding data use and privacy. Here, we wanted to highlight the performance of a range of NLP techniques, and were fortunate to have a rich, de-identified dataset to do so.
>
> Regarding your question about models using logistic / cross-entropy loss, yes, both our method and PPI++ in general are designed to be agnostic to how the prediction rule was trained and uses data-adaptive weights to incorporate information from ‘black box’ predictions into the overall procedure. We will make this more explicit in the main text.
>
> Lastly, we agree that including threshold-free metrics such as the ROC curve, ROC-AUC, PR curve, and PR-AUC would provide a more comprehensive evaluation of how our various NLP models perform. We will gladly revise Section 4.2 to include these metrics, as well as provide a more robust discussion on the performance of the method in light of these changes. Thank you for all of these valuable suggestions.

---

> > ### Comment · Reviewer_EkdW · 2024-06-04
> > **Re: Rebuttal by Authors**
> >
> > Thanks for your rebuttal. I think the methodology is not perfect, but the problem is important and the substantive application has potentially high impact.
> >
> > > We will revise the ethics statement
> >
> > > we agree that including threshold-free metrics
> >
> > Both these acknowledged revisions could also be included in-line in the response (if possible); I think this would be more compelling to the review team than a "commitment to revise upon acceptance".

---

> > > ### Author Response · Authors · 2024-06-07
> > > **Regarding your follow up to our rebuttal.**
> > >
> > > Thank you for this additional feedback. Given the additional space offered during the discussion period we will include more elaboration on our intended revisions.
> > >
> > > In the ethics statement in part B of the appendix we will add the following.
> > >
> > > "Data concerns are always paramount. We are fortunate to be working with de-identified data, however, when working with more sensitive data there are a number of open-source LLMs which can be run locally (Llama-3, meditron, etc) to ensure safe and ethical research practices. Additionally, many institutions, public and private, buy access to local OpenAI servers. Our use of these methods is illustrative, aiming to lay a foundational understanding of how to perform valid inference with NLP predictions."
> > >
> > > In our section 4.2 where we discuss our NLP prediction results, we will include a figure displaying the ROC, ROC-AUC, PR, and PR-AUC curves for the GPT-4 model performance.

---

### Decision · Program_Chairs · 2024-07-10

**Decision:**

Accept

**Comment:**

This work focusses on inferring numerical attributes of interest from free-text, ultimately to identify patterns from verbal autopsies. This is a shift, then, from the usual emphasis on *prediction* and toward *inference* over free-text. The work is clearly presented and there are clear contributions here.

While some reviewers raised concerns regarding evaluation, there was also broad consensus that this work constitutes an interesting direction and operationalizes this in service of an important problem in public health. The approach may be adopted for a range of related inference problems in public health.

[At least one review was discounted during the decision process due to quality]